# Deciphering photocarrier dynamics for tuneable high-performance perovskite-organic semiconductor heterojunction phototransistors

Yen-Hung Lin [1,10]*, Wentao Huang[2,10], Pichaya Pattanasattayavong [3], Jongchul Lim [1], Ruipeng Li[4], Nobuya Sakai[1], Julianna Panidi [2], Min Ji Hong[5], Chun Ma[6], Nini Wei[7], Nimer Wehbe[7], Zhuping Fei[8,9], Martin Heeney [8], John G. Labram [5], Thomas D. Anthopoulos [6]* & Henry J. Snaith [1]*

Looking beyond energy harvesting, metal-halide perovskites offer great opportunities to revolutionise large-area photodetection technologies due to their high absorption coefficients, long diffusion lengths, low trap densities and simple processability. However, successful extraction of photocarriers from perovskites and their conversion to electrical signals remain challenging due to the interdependency of photogain and dark current density. Here we report hybrid hetero-phototransistors by integrating perovskites with organic semiconductor transistor channels to form either "straddling-gap" type-I or "staggered-gap" type-II heterojunctions. Our results show that gradual transforming from type-II to type-I heterojunctions leads to increasing and tuneable photoresponsivity with high photogain. Importantly, with a preferential edge-on molecular orientation, the type-I heterostructure results in efficient photocarrier cycling through the channel. Additionally, we propose the use of a photo-inverter circuitry to assess the phototransistors' functionality and amplification. Our study provides important insights into photocarrier dynamics and can help realise advanced device designs with "on-demand" optoelectronic properties.

[1] Clarendon Laboratory, Department of Physics, University of Oxford, Parks Road, Oxford OX1 3PU, UK. [2] Department of Physics and Centre for Plastic Electronics, Imperial College London, London SW7 2AZ, UK. [3] Department of Materials Science and Engineering, School of Molecular Science and Engineering, Vidyasirimedhi Institute of Science and Technology (VISTEC), Rayong 21210, Thailand. [4] National Synchrotron Light Source II, Brookhaven National Laboratory, Upton, New York 11973, USA. [5] School of Electrical Engineering and Computer Science, Oregon State University, Corvallis, OR 97331, USA. [6] Physical Sciences and Engineering Division, King Abdullah University of Science and Technology (KAUST), Thuwal 23955-6900, Saudi Arabia. [7] King Abdullah University of Science and Technology (KAUST), Core Labs, Thuwal 23955-6900, Saudi Arabia. [8] Department of Chemistry and Centre for Plastic Electronics, Imperial College London, London SW7 2AZ, UK. [9] Institute of Molecular Plus, Tianjin Key Laboratory of Molecular Optoelectronic Science, Tianjin University, Tianjin 300072, China. [10] These authors contributed equally: Yen-Hung Lin, Wentao Huang. *email: yen-hung.lin@physics.ox.ac.uk; thomas.anthopoulos@kaust.edu.sa; henry.snaith@physics.ox.ac.uk

After several decades of research, the technological development for photodetection and photosensing have met the needs of a wide range of applications in everyday lives, including optical communications[1], imaging systems[2], environmental detection[3] as well as medical diagnostics[4]. Among different types of photodetectors, phototransistors are of notable interest due to their potential to achieve highly tuneable photogain with a variety of new intrinsic semiconductors[5,6]. Being an active component in detection, phototransistors are not only capable of operating under different modes but also able to reduce the complexity of the readout circuitry. Moreover, phototransistor-based approaches have extensively enriched the field of photodetection with noteworthy state-of-the-art demonstrations such as enhanced infrared detection for Si electronics by colloidal quantum dots[7], wide-dynamic-range photocarrier integration[8], and advanced imaging systems by combining with CMOS technologies[9,10].

Metal-halide perovskites (MHPs) stand out as obvious candidates for implementation in these devices, owing to their much-heralded material property trio: remarkable absorption coefficients[11], long-lived photocarriers[12] and tuneable optical bandgaps[13]. Together with simple processing requirements, this has led to their rapid emergence over the past years, not only promising to revolutionise future energy generation[14,15] but also for innovating optoelectronic applications[6,16,17]. In spite of their promising features, the phototransistors based on process-friendly semiconducting MHPs (i.e., solution-grown polycrystalline films) have not been widely reported[18]. The reason could be attributed to inter-grain boundary scattering[19] and/or intra-grain defect recombination[20]. Furthermore, several groups have reported that the ionic nature of MHPs appears to cause difficulties for gating the transistor channel at room temperature[21,22]. Research efforts have since been put into devices based on single crystals[23], single-crystalline layers[24] and monolayer flakes[25]. However, the rather complex process protocols to form single-crystalline materials are less scalable, hence posing a great challenge for high-throughput, large-area manufacturing[17]. On the other hand, hybrid heterostructures that combine materials with complementary optical and electrical properties in a phototransistor serve as an excellent platform to enhance the photoresponse[5]. Numerous attempts using MHPs in this manner have been reported (see Supplementary Note 1 and Supplementary Fig. 1), but their mediocre optoelectronic performance is usually mitigated by employing a highly conductive channel material, resulting in a photoconductor, rather than a phototransistor, with an apparent trade-off between photogain and dark current levels.

Here, we systematically investigate charge transfer and photocarrier transport properties based on a conceptual heterojunction phototransistor (HJPT) architecture, employing the mixed-cation mixed-anion perovskite, $FA_{0.83}Cs_{0.17}PbI_{2.7}Br_{0.3}$ (FACs, FA = formamidinium)[13] as the light absorber material and two types of high-mobility p-type organic semiconductors (OSCs) {the small molecule 2,7-dioctyl[1]benzothieno [3,2-b][1]benzothiophene ($C_8$-BTBT)[26] and the conjugated polymer indacenodithiophene-benzothiadiazole ($C_{16}$-IDTBT)[27]}, as well as their blends, as the conducting channel materials. Using a range of experimental characterisation techniques, we decipher the dynamics behind photocarriers' charge transfer process and photoresponse mechanisms in our hybrid phototransistors. We attribute the enhanced photoresponse to the careful engineering of a "straddling gap" type-I heterojunction between the MHP and OSC, rather than employing an intuitive, energetically-favourable "staggered gap" type-II heterojunction[8]. In our type-I HJPT, the long-lived photocarriers in the MHP can propagate along the OSC with minimum recombination losses, resulting in the enhanced photoresponse. We believe that this concept is not limited to the

materials used in this work; rather, it could be extended to resolve undesired interfacial losses in other types of hybrid phototransistors. Furthermore, we fabricated unipolar photo-inverters using phototransistors, demonstrating photo-modulated output voltages and evaluating their intrinsic amplification. Not only will the knowledge acquired from our work help to provide tailorable, on-demand optoelectronic sensors, but the large library of MHP and OSC materials could also address many other optoelectronic applications, by applying this unique type-I HJ approach to facilitate better photocarrier transport.

## Results

**HJPTs and performance metrics**. Our HJPT consisting of $C_8$-BTBT and $C_{16}$-IDTBT OSCs (Fig. 1a) in a top-gate bottom-contact (TG-BC) transistor configuration is shown in the schematic drawing in Fig. 1b. The design of our HJPT sandwiches the light-absorbing layer MHP between the source-drain (S-D) contacts and the OSC transistor channel. Figure 1c shows the ultraviolet–visible (UV–vis) absorption spectra for the films of FACs, $C_8$-BTBT, $C_{16}$-IDTBT and a blend composed of $C_8$-BTBT and $C_{16}$-IDTBT in a ratio of 1:1 (abbreviated as blend 1:1). The absorption edge for FACs appears slightly below 800 nm whilst those of $C_8$-BTBT and $C_{16}$-IDTBT/blend 1:1 start at ~400 nm and ~730 nm, respectively. Since $C_8$-BTBT and $C_{16}$-IDTBT possess different highest occupied molecular orbital (HOMO) levels (5.5 eV for $C_8$-BTBT[28] and 5.1 eV for $C_{16}$-IDTBT[29,30]) and the valence band (VB) for FACs is around 5.4 eV[13], the OSC materials chosen to construct HJPTs could form either straddling-gap type-I heterojunctions ($C_8$-BTBT) or staggered-gap type-II heterojunctions ($C_{16}$-IDTBT)[31] with FACs (see Fig. 1d). Figure 1e shows surface topographic images acquired from atomic force microscopy (AFM) for the hetero FACs/OSC bilayer stacks and the pristine FACs film with the corresponding height distribution presented in Fig. 1f. Apart from the clear grain-like surface features (Fig. 1e), the pristine FACs film exhibits two distinct peaks in its height distribution (Fig. 1f). This is attributed to the co-existence of thick perovskite-rich areas as well as thin perovskite-deficient regions in the FACs film. On the other hand, for all the FACs/OSC bilayer stacks we only observed one Gaussian-like distribution regardless of what OSC compositions were used. Therefore, in the bilayer stacks the OSCs formed conformal coatings with good coverage over the grainy FACs layer beneath. To further inspect the FACs/OSC bilayer stacks, we carried out the high-angle annular dark-field scanning transmission electron microscopy (HAADF-STEM) characterisation on the FACs/blend 1:1 film stack, and the images are shown in Supplementary Fig. 2. The film thicknesses obtained for the FACs and blend 1:1 layers are 30~ 50 nm and 30~ 40 nm, respectively. Such a continuous channel layer is critical for long-range carrier-transport pathways in laterally-structured devices like transistors[32].

Figure 1g shows the GIWAXS patterns of the pristine OSC films and the FACs/OSC bilayers, which resemble the actual film stack in our perovskite HJPTs. One X-ray incident angle of 0.1° was used for studying the pristine OSC films (Fig. 1g) whilst for the bilayer stacks, two different incident X-ray angles of 0.1° (Fig. 1g) and 0.05° (Supplementary Fig. 3) were carried out in the GIWAXS measurements. From Fig. 1g, the FACs layer exhibits ring-like features at higher $q$ space, but they are only observable at high (0.1°), not low (0.05°), incident angles. When the incident angle is as low as 0.05°, the X-ray with an energy of 13.5 keV mainly probes the top surface of the target films whilst at the higher incident angle of 0.1°, it can penetrate the top OSC layer and probe the underlying FACs layer. The GIWAXS pattern for the pristine $C_8$-BTBT film shows the (001) peak at $q = 0.25$ Å$^{-1}$ and a series of (00L) peaks along the out-of-plane ($q_z$) direction

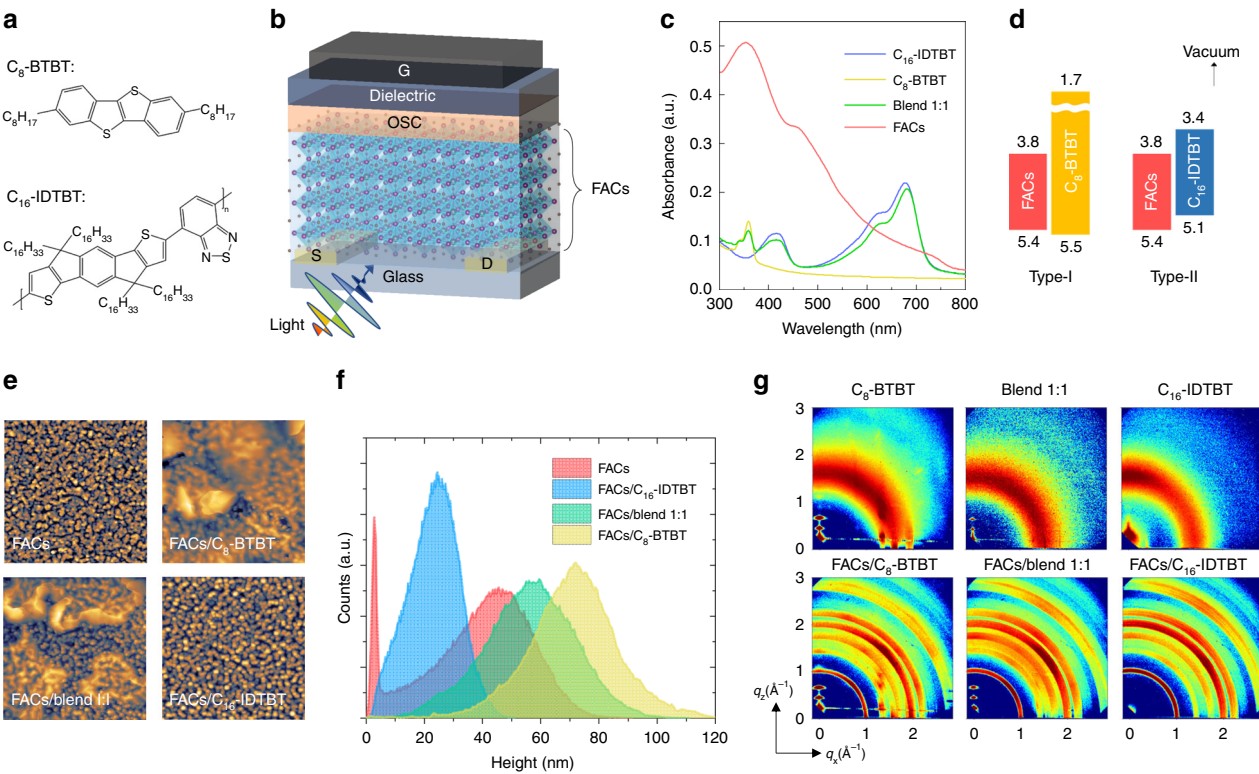

**Fig. 1** Perovskite heterojunction phototransistors. **a** Chemical structures of a small molecule $C_8$-BTBT and a copolymer $C_{16}$-IDTBT. **b** Schematic of HJPT configured in a TG-BC architecture, where G, S and D denote the gate, source and drain electrodes, respectively. An MHP/organic semiconductor (OSC) heterostructure ($FA_{0.83}Cs_{0.17}PbI_{2.7}Br_{0.3}$, i.e. FACs, as the MHP layer) is employed as a dual function photoresponsive transistor channel. Incident light absorbed by the FACs layer can trigger the photodetection functionality in HJPTs. **c** Ultraviolet–visible (UV-vis) absorption spectrum for $C_{16}$-IDTBT, $C_8$-BTBT, an organic blend of $C_8$-BTBT:$C_{16}$-IDTBT in a ratio of 1:1 (blend 1:1) and FACs. **d** Energy (in eV) band alignments of type-I and type-II heterojunctions for FACs/$C_8$-BTBT and FACs/$C_{16}$-IDTBT, respectively. **e** Representative AFM topographic images (5 μm × 5 μm) for different MHP/OSC bilayer stacks: FACs; FACs/$C_8$-BTBT; FACs/blend 1:1; FACs/ $C_{16}$-IDTBT. **f** Corresponding height distributions derived from the AFM results shown in (**e**). **g** GIWAXS patterns for the pristine OSC films and FACs/OSC bilayer stacks using an X-ray incident angle of 0.10°

($L = 1, 2, 3…$), presenting a typical edge-on orientation[33]. This suggests that the $C_8$-BTBT molecules stack along the planar directions, allowing for efficient in-plane, lateral charge transport, suitable for transistor applications. On the other hand, the pristine $C_{16}$-IDTBT thin film shows both (200) peak at $q = 0.48 Å^{-1}$ and (010) peak at $q = 1.65 Å^{-1}$ along the $q_z$ direction, corresponding to the backbone stacking and π-π stacking of the polymer chain, respectively. This result reveals the coexistence of edge-on and face-on orientations in the pristine $C_{16}$-IDTBT film. The blend 1:1 film shares a similar structure with the pristine $C_8$-BTBT, suggesting the same orientation for $C_8$-BTBT in the blend and $C_{16}$-IDTBT being amorphous. As shown in both Fig. 1g and Supplementary Fig. 3, the FACs/$C_8$-BTBT and FACs/blend 1:1 bilayer stacks share similar scattering features with the corresponding pristine OSC films. The GIWAXS patterns do not exhibit any pronounced OSC structural features in the FACs/$C_{16}$-IDTBT sample, indicating that $C_{16}$-IDTBT forms an amorphous-like layer on top of FACs.

To examine the optoelectronic properties of our HJPTs, we fabricated five types of FACs HJPTs with different OSC compositions, including pure $C_8$-BTBT (1:0), pure $C_{16}$-IDTBT (0:1), and three $C_8$-BTBT:$C_{16}$-IDTBT blends with ratios of 3:1, 1:1 and 1:3. Figure 2a shows the current–voltage (I-V) characteristics for FACs/OSC HJPTs with different OSC compositions operated at a constant drain voltage ($V_D$) of −40 V and under different illumination intensities from a blue light LED ($λ_{peak} = 475$ nm). Figure 2b–d present the gate bias-dependent photoresponsivity for the HJPTs under the

illumination of three LED light sources with incident optical power density of ~7 μW cm$^{-2}$ and spectral peaks centred at 632 nm (Fig. 2b), 525 nm (Fig. 2c) and 475 nm (Fig. 2d). The original current–voltage (I-V) characteristics collected in a dark environment and under different illumination light sources are shown in Supplementary Fig. 4–8 and discussed in Supplementary Note 2, along with additional I-V data taken under higher optical power densities (~70 μW cm$^{-2}$ and ~700 μW cm$^{-2}$). A photoresponsivity of $>10^4$ A W$^{-1}$ appears to be readily achievable with most of our HJPT configurations. Among different OSC compositions, the $C_8$-BTBT based HJPT exhibits the highest photoresponsivity of $2 × 10^4$ A W$^{-1}$. Yet, the most noticeable feature for our HJPTs is the photoresponsivity measured at/near the transistors' off-state (i.e., low dark-current levels) that exhibits significant differences from one another. Taking the photoresponsivities obtained from the $C_8$-BTBT and the blend 1:1 devices as examples, the $C_8$-BTBT HJPT appears to conduct a large number of photocarriers as compared to the others when operated in the regime of low dark current levels;[8] the blend 1:1 HJPT on the other hand, resembles the I-V characteristics of a near-ideal hybrid phototransistor (as discussed in ref.[5]) with high photoresponse around the turn-on voltage ($V_{ON}$) and suppressed photoinduced channel current at the transistor's off-state[5]. In other words, adjusting the $C_8$-BTBT: $C_{16}$-IDTBT blend composition can deliver tuneable, 'on-demand'-like photoresponsivity using our HJPT approach, extending the HJPT's suitability for a variety of optoelectronic applications[7–9]. The photoresponsivity achieved in our HJPT approach is among the highest reported to

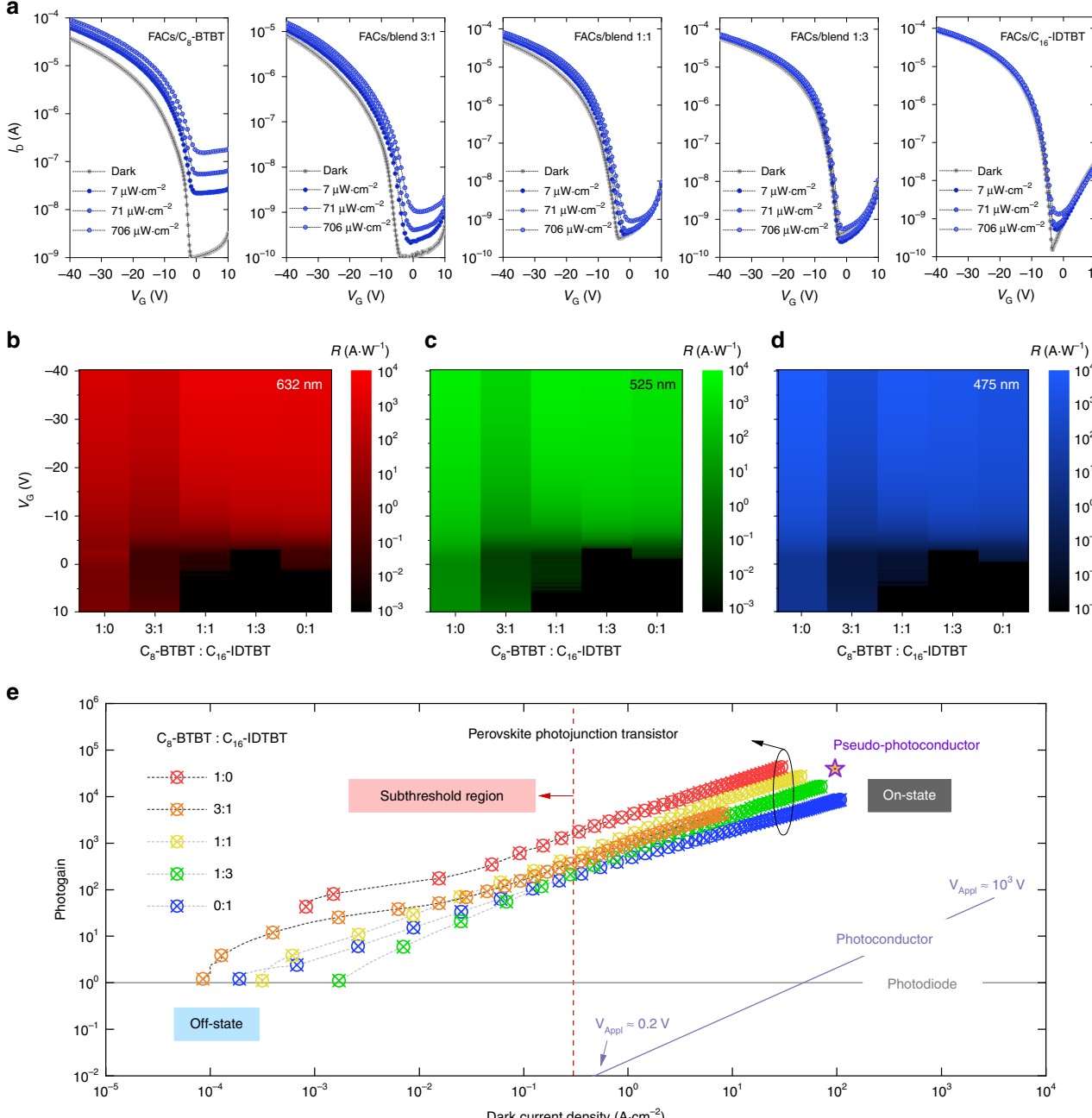

**Fig. 2** On-demand photoresponsivity and photogain analysis. **a** Transfer I-V characteristics ($V_D = -40$ V) for FACs/OSC HJPTs based on pristine $C_8$-BTBT, pristine $C_{16}$-IDTBT and organic blends with different $C_8$-BTBT:$C_{16}$-IDTBT blend ratios under a 475-nm LED light source with various intensities. **b–d** Gate-voltage dependent photoresponsivity (R) measured at $V_D = -40$ V from HJPTs based on $C_8$-BTBT, $C_{16}$-IDTBT and different blend ratios under different light sources: **b** 632 nm; **c** 525 nm; **d** 475 nm, all under ~7 $\mu$W cm$^{-2}$ incident optical power density. **e** Photogain as a function of dark current density for HJPTs with different OSCs at ~ 7 $\mu$W cm$^{-2}$ incident power from a 475-nm LED light source. For comparison, a photoconductor model is calculated based on the same channel dimension using the drift current equation[31] with applied voltages ($V_{appl}$) ranging from 0.2 to $10^3$ V. A pseudo-photoconductor modelled to achieve a high-gain level (~$4 \times 10^4$) is set to operate at $V_{appl} = 40$ V using the same parameters for the drift current equation, but requires a carrier lifetime of 20 ms. The theoretical limit for the photogain from a photodiode is drawn for reference

date for MHP-based phototransistors (see Supplementary Fig. 1). Since the HJPTs with $C_8$-BTBT and/or $C_{16}$-IDTBT can only deliver field-effect mobility < 1 cm$^2$ V$^{-1}$ s$^{-1}$ (Supplementary Table 1), the reason for this outstanding performance cannot be attributed to an excessive carrier density. The latter is a characteristic of a photoconductor, rather than a phototransistor, and results in the disadvantageous high dark current densities, as discussed in detail in Supplementary Note 1.

In Fig. 2e, we plot the relation between photogain and dark current density for our HJPTs as well as simulated photodetectors and photoconductors based on the intuitive model from the drift current equation[7,31]. For photoconductors, the simulation was carried out using the same device dimensions as our HJPT (see "Methods") and the following parameters that resemble MHP-like properties:[21,22] carrier lifetime of 1 $\mu$s, carrier concentration of $10^{17}$ cm$^{-3}$, mobility of 0.6 cm$^2$ V$^{-1}$ s$^{-1}$, film thickness of 500 nm,

and applied voltages ($V_{appl}$) ranging from 0.2 to 1000 V. Conventional photodiodes on the other hand, being a passive element, could only generate a maximum photogain of 1 under the condition of reaching an external quantum efficiency of 100%[31]. Regarding the plots of our HJPTs in Fig. 2e, the dark current densities were measured under different applied gate voltage ($V_G$) and at a fixed $V_D = -40$ V whilst the illumination was carried out with a light source of ~7 μW cm$^{-2}$ at 475 nm. Evidently, our HJPT approach can result in high photogain (with the highest photogain reaching ~$4.5 \times 10^4$) at low dark current levels[7]. Such a result is comparable to the state-of-the-art hybrid-type phototransistors[7] and surpasses the simulated single-component MHP photoconductor model (light purple line shown in Fig. 2e) by several orders of magnitude. Ideally, if a pseudo-photoconductor is set to operate at $V_{appl} = 40$ V with the same aforementioned parameters, it will require an extremely long carrier lifetime of ~20 ms in order to achieve such a high photogain of $4 \times 10^4$. Moreover, a noticeable feature, which appears between the off-state and the subthreshold region, is that the devices based on $C_8$-BTBT and 3:1 blend explicitly exhibit photogain plateau, i.e., slow decrease in photogain. However, when operating towards the high-gain region, both follow a power-law increase, similar to the HJPTs based on other OSC compositions. As such, for those devices based on $C_8$-BTBT and 3:1 blend the photogain levels are maintained even when the dark current density starts decreasing as the transistor further enters the deep-subthreshold operation (i.e., close to off-state).

**Spectroscopic analysis of photoinduced charge carriers**. The most peculiar characteristic in our HJPTs is their tuneable photoresponsivity at/near the transistor off-states. This operation regime has no or little gate field to attract photoinduced carriers to cross the FACs/OSC heterojunctions. Therefore, we carried out a series of spectroscopic studies, which do not involve any direct out-of-plane field-effect, on the FACs/OSC bilayer stacks in order to elucidate the driving force behind this tuneable photoresponse characteristic. Figure 3a, b shows the results of FACs/$C_8$-BTBT, FACs/blend 1:1 and FACs/$C_{16}$-IDTBT obtained from time-resolved and steady-state photoluminescence (TRPL and SSPL) studies, respectively. We observed that the PL signal from the FACs/$C_8$-BTBT is similar to that from the pristine FACs sample. This can be explained based on the type-I heterojunction (Fig. 1d), which preserves the majority of the photocarriers. On the other hand, the PL signal from the FACs/$C_{16}$-IDTBT sample is short-lived and strongly quenched, pointing that the photocarriers in FACs can be effectively transferred out of FACs, which agrees with the type-II heterojunction configuration (Fig. 1d). The TRPL and SSPL results strongly confirm our hypothesis of the heterojunction types formed by the different OSCs. As for the FACs/blend 1:1 sample, its PL characteristics are similar to those of FACs/$C_{16}$-IDTBT, suggesting that the OSC material in the blend which is in direct contact with the FACs is mostly $C_{16}$-IDTBT. These observations in fact, are in line with the pronounced vertical phase separation for small molecule and polymer blends reported previously[30,32], where they show that $C_8$-BTBT mostly forms in the upper part of a blend layer when mixed with $C_{16}$-IDTBT. To gain more direct evidence, we performed the dynamic secondary ion mass spectrometry (DSIMS) characterisation on the FACs/blend 1:1 film stack, and the result is shown in Supplementary Fig. 9. It is clear that in the first 10 nm, we observed a region lacking CN$^-$ ions that indicates only a small fraction of (or barely any) $C_{16}$-IDTBT. On the other hand, larger counts for the CN$^-$ ions are observed close to the region consisting of rich perovskite-related species (I$^-$, Br$^-$ and PbI$^-$). As such, we can confirm that the upper part of a $C_8$-BTBT: $C_{16}$-

IDTBT blend is dominated by $C_8$-BTBT whilst the lower part of the blend is a region that $C_{16}$-IDTBT populates.

To further probe the nature of charge-transfer between the FACs and OSC layers, their bilayer stacks were studied using time-resolved microwave conductivity (TRMC, Supplementary Note 3, Supplementary Figs. 10–12 and Supplementary Table 2). TRMC enables one to evaluate a proxy for the sum of intrinsic electron and hole mobilities, without the need to form a device or deposit electrical contacts[34,35]. Figure 3c shows the TRMC figure of merit $\phi\Sigma\mu_{TRMC} = \phi(\mu_e + \mu_h)$ for pristine FACs and FACs/OSC bilayer stacks as a function of optical fluence, where $\phi$ is the charge-generation efficiency, and $\mu_e$ and $\mu_h$ are the electron and hole mobilities, respectively. At high optical fluence, a significant amount of bimolecular and Auger recombination occurs during the ~5 ns laser pulse, resulting in a reduction in the peak observable photoconductance, and hence a decrease in the extracted $\phi\Sigma\mu_{TRMC}$ (Fig. 3c)[34,36]. A model that approximates representative values of $\phi\Sigma\mu_{TRMC}$ is applied to our experimental data[37]. From these fits, we evaluate $\phi\Sigma\mu_{TRMC} = 0.8$ cm$^2$ V$^{-1}$ s$^{-1}$ for the pristine FACs film, $\phi\Sigma\mu_{TRMC} = 0.61$, 0.15 and 0.2 cm$^2$ V$^{-1}$ s$^{-1}$ for the FACs/$C_8$-BTBT, FACs/$C_{16}$-IDTBT and FACs/blend 1:1 bilayer stacks, respectively. Both the pristine FACs and FACs/$C_8$-BTBT bilayer have an approximately three to fourfold increase in $\phi\Sigma\mu_{TRMC}$ as compared to the FACs/$C_{16}$-IDTBT and FACs/blend 1:1 bilayer stacks.

$\phi\Sigma\mu_{TRMC}$ as determined by the TRMC technique is representative of the local sum of electron and hole mobilities, in the plane of the substrate, averaged over the whole sample. In an attempt to draw an analogy between the spectroscopic analysis and HJPT operation from the viewpoint of in-plane charge transport, we carried out transient photoconductivity (TPC) measurement to determine long-range lateral mobility $\phi\Sigma\mu_{TPC}$[38,39]. Our TPC study (Fig. 3d) was carried out using a pulsed laser with a range of excitation densities to illuminate the pristine FACs and FACs/ OSC bilayer stacks made on the glass substrates with two predeposited in-plane Au electrodes. The schematic drawing of the experimental setup is shown in Supplementary Fig. 13. Knowing the photoconductivity as a function of the laser excitation density (Supplementary Fig. 14), we estimated the long-range sum of electron and hole mobility within the FACs films using a simple relationship reported previously, i.e., $\sigma_{TPC} = eI_{abs}\phi\Sigma\mu_{TPC}$ where $e$ is elementary charge, $I_{abs}$ is absorbed excitation density as determined from the laser pulse intensity and the absorbance of the film[38]. As such, we were able to determine the long-range lateral mobility $\Sigma\mu_{TPC}$ to be 0.69, 0.52, 0.19 and 0.22 cm$^2$ V$^{-1}$ s$^{-1}$ (at the lowest excitation density of $1.57 \times 10^{17}$ cm$^{-3}$) for the pristine FACs, FACs/$C_8$-BTBT, FACs/$C_{16}$-IDTBT and FACs/ blend 1:1, respectively. Notably, these long-range mobilities of the pristine FACs and FACs/OSC bilayers are in good agreement with the average local mobilities determined by TRMC.

To summarise the spectroscopic analysis presented in Figs. 3a–d, we consider the results as convincing evidence that upon illumination, photoinduced charges are readily transferred from FACs to $C_{16}$-IDTBT or a $C_8$-BTBT:$C_{16}$-IDTBT blend, but less so from FACs to $C_8$-BTBT. In making this claim, we assume that the sum of charge carrier mobilities obtained from our spectroscopic measurements is higher in the FAC film than those in the OSCs[40], hence allowing us to interpret a lower measured mobility being representative of a greater proportion of carriers in the OSC vs. the FACs film. We surmise that the observed difference in charge-transfer is due to the energetic levels between FACs and OSCs giving rise to different HJ configurations (Fig. 1d). Also, the differences in morphology at the interface between the OSC and FAC could play a role, i.e., $C_8$-BTBT molecules were observed from our GIWAXS data (Fig. 1g) to

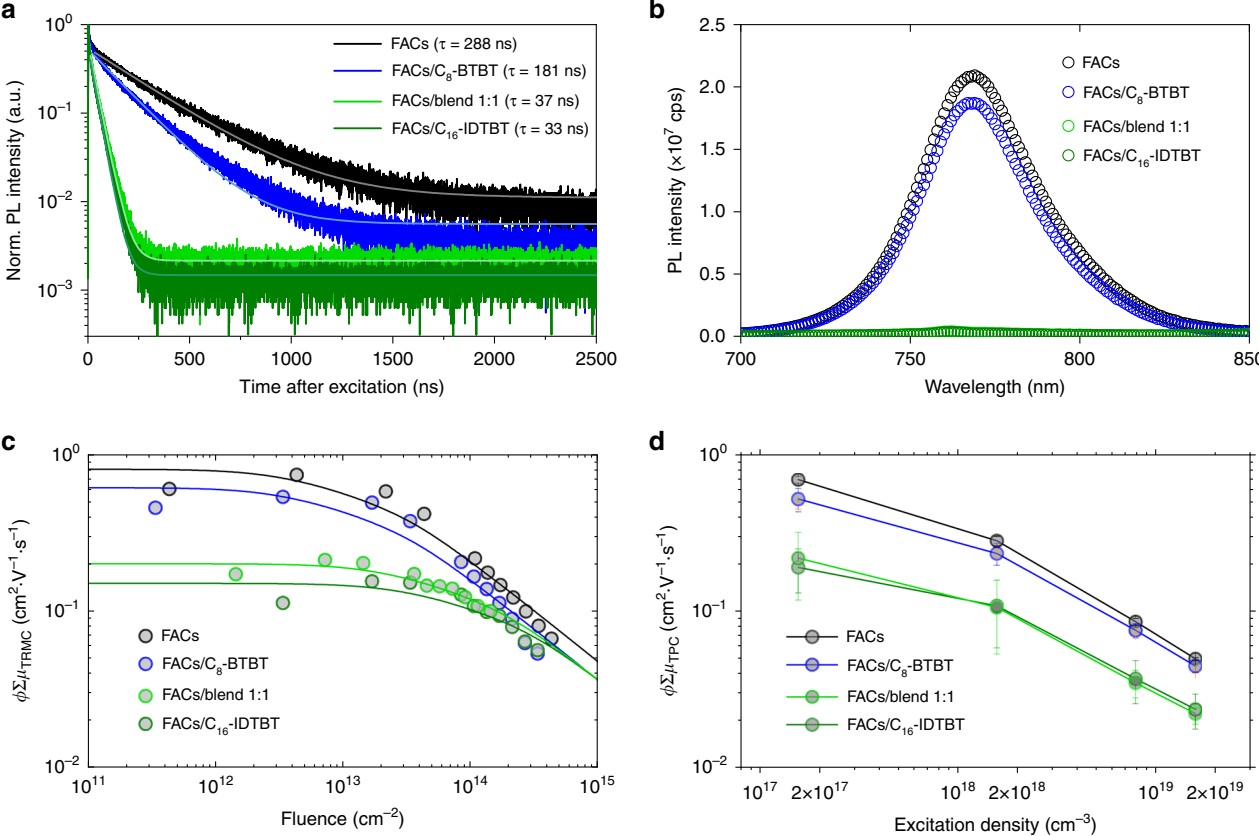

**Fig. 3** Spectroscopic characterisations on photocarriers transfer dynamics. **a**, **b** Photoluminescence (PL) spectroscopy results for FACs and FACs/OSC bilayer stacks: **a** time-resolved PL (TRPL) measurements and **b** steady-state PL (SSPL) measurements. In (**a**) the time constant $\tau$ for each sample was extracted using the rate based on a mono-exponential decay. **c** Time resolved microwave conductivity (TRMC) figure of merit $\phi\Sigma\mu_{TRMC}=\phi(\mu_e+\mu_h)$, plotted as a function of incident optical fluence for FACs and FACs/OSC bilayer stacks on quartz substrates. The points are experimental data, and the lines are fits to a numerical model[37], which accounts for bimolecular and Auger recombination at high fluence. **d** Transient photoconductivity long-range lateral mobility $\phi\Sigma\mu_{TPC}$, plotted as a function of excitation density by varying incident laser fluences for FACs and FACs/OSCs bilayer stacks. The error bars denote standard deviations

form an edge-on orientation. The latter is not as beneficial as face-on orientation to help vertical charge transfer, as widely reported in the field of organic photovoltaics[41].

**Mott-Schottky analysis and perovskite-organic diodes**. To elucidate the relationship between efficient circulation of photocarriers and high photoresponse, and hence high photogain, we carried out the Mott-Schottky analysis on the FACs/OSCs bilayer stacks. This is in order to gain insight into how the photoinduced charge profiles are distributed in the proximity of various FACs/OSCs heterointerfaces in the dark environment and upon light illumination. Meanwhile, we fabricated diodes by sandwiching three different FACs/OSC bilayer stacks between Au and Au electrodes to interrogate their optoelectronic characteristics (Supplementary Note 4 and Supplementary Fig. 15).

To enhance the transparency for illumination from the glass side, the experimental film stacks for the Mott-Schottky analysis were made on ITO-coated glass with the structure of glass/ITO/FACs/OSC/Au (Supplementary Fig. 16). Our analysis was carried out at an intermediate frequency of ~5 kHz to minimise the effects of series resistance as well as surface and interface dipoles[42–44]. Figure 4a, b, c shows the Mott-Schottky results for FACs/C$_{16}$-IDTBT, FACs/blend 1:1 and FACs/C$_8$-BTBT, respectively. In the dark, both the FACs/C$_{16}$-IDTBT and FACs/blend 1:1 samples exhibit two distinct regions in their Mott-Schottky plots (Fig. 4a, b): a transition region with an uprising trend (going

from forward to reverse bias) and flat regions with constant capacitance levels. For the former, this is the operation regime where charge depletion/accumulation takes place whilst the latter indicates combined geometric capacitance from bulk dielectric polarisation of the FACs/OSC bilayer stacks[43]. For the FACs/C$_8$-BTBT sample (Fig. 4c), only one distinct region with a constant capacitance level is observed during the entire operational regime. This characteristic indicates that the FACs/C$_8$-BTBT bilayer forms a dielectric stack. Such a result is in line with the electrical characterisation obtained from our attempt to make a FACs/C$_8$-BTBT diode (Supplementary Fig. 15c).

However, under illumination, the FACs/C$_{16}$-IDTBT (Fig. 4a) and FACs/blend 1:1 (Fig. 4b) samples exhibit charge polarity inversion behaviour with a depletion region for p-type charge carriers at low forward bias corresponding to the decrease in the capacitance (i.e., higher C$^{-2}$). This is followed by the accumulation of the oppositely charged carriers (i.e. n-type) associated with the increase in the capacitance. Under large reverse bias, the Mott-Schottky plots for both FACs/C$_{16}$-IDTBT and FACs/blend 1:1 returns to a high capacitance level. For the latter behaviour, the true enhancement of the bulk dielectric polarisation in FACs still needs to be clarified. This is because to show that such enhancement is taking place, one needs to exclude the effects caused by the accumulation of surface/interface dipoles, which also result in a high capacitance level under light[43]. Nevertheless, the clear charge polarity inversion when entering the reverse bias regime (Fig. 4a, b) is a direct indication that negatively charged

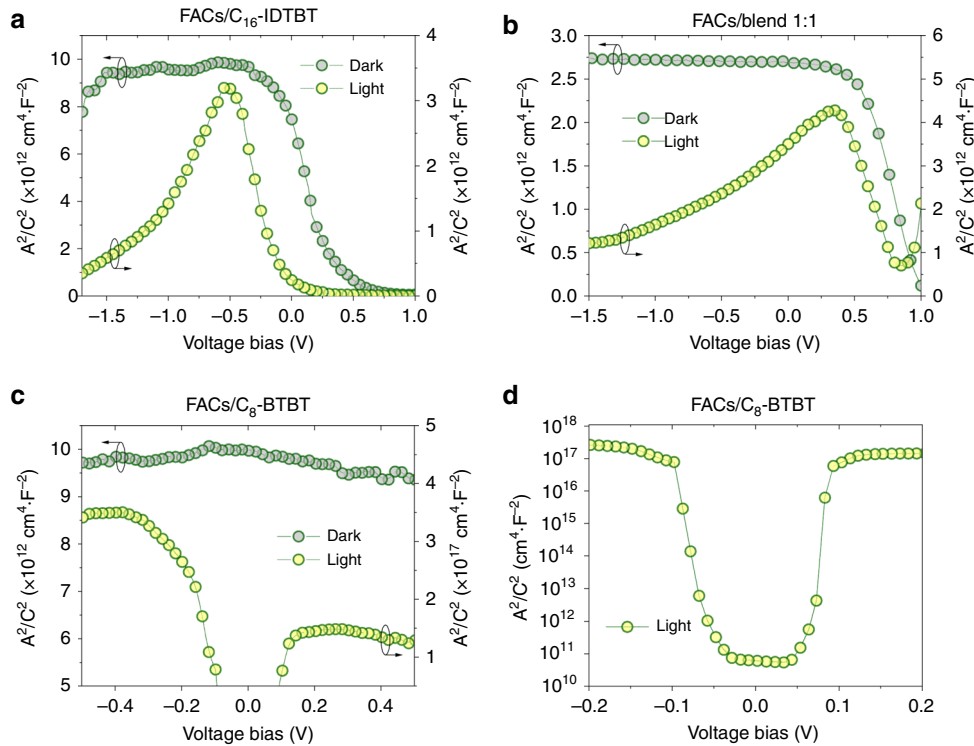

**Fig. 4** Mott-Schottky analysis for FACs/OSC bilayers. **a–d** Mott-Schottky plots for different FACs/OSC bilayer stacks: **a** FACs/$C_{16}$-IDTBT; **b** FACs/blend 1:1; **c** FACs/$C_8$-BTBT; **d** shows the detailed changes for the voltage bias between −0.2 and 0.2 V in (**c**). In (**a**)–(**d**), C/A is geometric capacitance (F cm$^{-2}$) where C and A stand for capacitance (F) and area (cm$^2$), respectively. The impedance spectroscopy measurements were carried out with an AC perturbation of 10 mV at 5 kHz in the dark (grey-filled dark green circles) as well as under a white-light LED illumination source (~2.61 mW cm$^{-2}$; data shown in yellow-filled bright green circles)

carriers (i.e., electrons) are left behind after positively charged carriers (i.e., holes) are transferred to the OSC layers. This result is supported by our spectroscopic data shown in Fig. 3. On the other hand, in Fig. 4c upon illumination the FACs/$C_8$-BTBT sample exhibits high capacitance levels measured under very low bias conditions (see Fig. 4d, from −0.1 to 0.1 V), but outside this regime, extremely low capacitance was measured. For the latter, the most plausible reason being that this bilayer dielectric stack could not withhold any charge carriers as a result of capacitance loss, due to the manifestation of series resistance effects. The two different capacitance levels outside ±0.1 V also correspond to different charge extraction efficiencies. This result agrees with the I-V characteristics of the FACs/$C_8$-BTBT diode (Supplementary Fig. 15c), i.e., higher current levels (corresponding to higher capacitance levels) at positive bias than negative bias.

To further clarify any ambiguity regarding the role OSCs play in the photocarrier transport, we processed the FACs-only phototransistor in order to shed more light on the impact of the use of the OSC transistor channel on the photodetection performance. Supplementary Fig. 17a compares the FACs-only device with two HJPTs using either $C_8$-BTBT or $C_{16}$-BTBT as the channel material. The FACs-only phototransistor exhibits minimal changes in photoresponse across the entire $V_G$ operation range, indicating that in this case the photoresponse is weakly $V_G$-dependent. The latter can be explained by the fact that the ions in the perovskite-only transistor could screen the applied gate bias[22]. In contrast, our HJPT approach employs the OSCs as the transistor channel in which the applied gate bias can effectively induce the channel drain current, hence resulting in $V_G$-dependent photodetection. Such $V_G$-dependent photoresponse is in fact the key that separates 'phototransistors' from other types of photodetectors and allows phototransistors to

achieve high photoresponsivity, as detailed in our Supplementary Note 1. The impact of using our HJPTs on photodetection can also be realised by plotting the relation between photogain and dark current density (Supplementary Fig. 17b). In particular, the formation of FACs/$C_8$-BTBT type-I HJPT not only exhibits excellent channel modulation but also leads to highly effective photocarrier circulation.

From the information presented so far, Fig. 5 illustrates the photocarrier transport mechanism in the most intriguing type-I HJPT using the FACs/$C_8$-BTBT film stack. The pronounced photoresponse at or near the off-state (i.e. operating under a weak gate-induced electric field) can be related to the observation of the largest increase of the capacitance at zero bias (Fig. 4d). The latter suggests strong formation of photogenerated holes accumulated at/near the FACs/$C_8$-BTBT interfaces[43] as depicted in Fig. 5. Such interpretation coincides with the spectroscopic data (Fig. 3) as upon light illumination, no significant charge transfer from FACs to $C_8$-BTBT was observed. The latter is attributed not only to the type-I heterojunction formation between FACs and $C_8$-BTBT (Fig. 1d), but also to the evident edge-on orientation of $C_8$-BTBT (Fig. 1g), which helps those photoinduced carriers to transport along the regions at/near the heterointerfaces of FACs and $C_8$-BTBT and manifested as photocurrent.

Regarding the low photocurrent in the FACs/$C_{16}$-IDTBT HJPT (Fig. 2a), our spectroscopic characterisation results (Fig. 3) indicate that there were indeed photogenerated holes transferred out of the FACs layer. We then attribute the loss of photogenerated holes to the nonradiative recombination processes at the FACs/OSC interfaces[45]. The latter is often considered one of the most critical factors that ultimately hinders the performance of perovskite photovoltaics[46,47]. Such loss could be due to numerous possible causes from the copolymer, such as

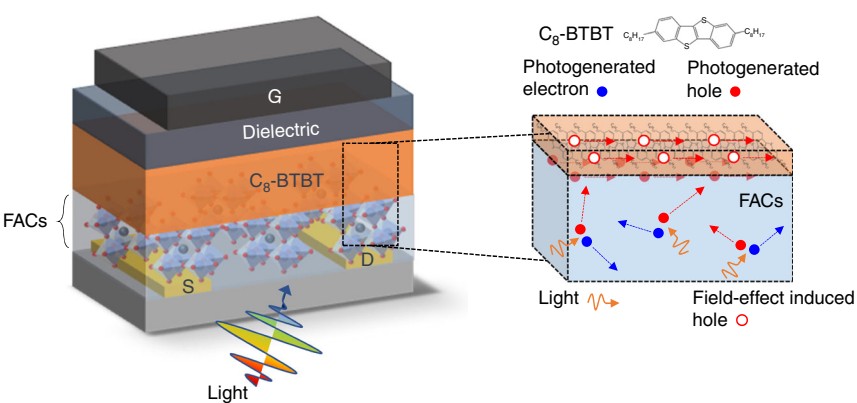

**Fig. 5** Operational mechanism for FACs/$C_8$-BTBT type-I HJPTs. Illustration of the photocarrier transport mechanism in the type-I HJPT using the FACs/$C_8$-BTBT film stack where G, S and D denote the gate, source and drain electrodes, respectively. The photogenerated holes excited by light illumination can diffuse and/or accumulate at/near the FACs/$C_8$-BTBT interfaces under the influence of a weak gate field-effect and transport along the $C_8$-BTBT transistor channel formed of organic molecules packed in an edge-on orientation, resulting in the enhanced photoresponse at/near the off-sate

the donor unit IDT, the acceptor unit BT, the side chain, or the bridging atoms, and adjusting these constituents could have an impact on the molecular packing and aggregation of the polymer backbone, hence resulting in different efficiencies for charge transport[48,49]. An in-depth and systematic investigation will be required to reveal and identify the origin of these nonradiative losses. On the other hand, two published works have showed that forming an insulating wide-bandgap layer at the interface between the perovskite and charge transporting layers could effectively reduce the interfacial nonradiative losses[50,51]. There is also an example that demonstrates how to tackle interfacial losses by adding a layer of electron-blocking $MoO_3$ in-between the light absorber and the phototransistor[52]. We note that even in a type-II HJ-based perovskite quantum-dot/organic phototransistor, the photocarrier transfer could still benefit from the insulating ligands formed on the surfaces of the quantum dots[53]. To this end, a type-II HJPT might be an intuitive way to facilitate the photocarrier transfer from the light absorber to the transistor channel; however, there still exist losses that could take place after the photocarriers leave the light absorber.

**Assessing performance metrics via a photo-inverter.** Key performance metrics[54], such as photoresponsivity (Supplementary Note 1), noise-equivalent power, specific detectivity (Supplementary Note 5, Supplementary Fig. 18 and Supplementary Table 3) and response time (Supplementary Note 6 and Supplementary Fig. 19) are traditionally defined for photodiodes. Since phototransistors combine detection and amplification into a single device, it is more informative to consider their utility as a circuit element. Here we propose a simple circuit route that uses a photo-inverter (Fig. 6a) to assess a phototransistor's performance metric in a manner more applicable for optoelectronic circuit design. To demonstrate this concept, a PMOS-like photo-inverter served as a common-source amplifier was fabricated using two identical FACs HJPTs, where HJPT-1 and HJPT-2 respectively, were employed as the depletion load and electrical switch controlled by the input voltage signal ($V_{IN}$). In this unipolar configuration, one can directly evaluate the intrinsic amplification ($A_i$) of the optoelectronic components (e.g. HJPT-1 and HJPT-2) for amplifying input signals[55] as the latter is the main advantage for using an active electronic component like phototransistors. Moreover, $A_i$ is an important criterion to separate a phototransistor from a photoconductor as a high $A_i$ requires the saturation of $I_D$ (Supplementary Note 7 and Supplementary Fig. 20).

In our demonstration, we focused on the operation regime that corresponds to the lower dark current levels. Fig. 6b, c shows the

photo-inverters' operations based on FACs/$C_{16}$-IDTBT and FACs/$C_8$-BTBT, respectively. The sharp voltage swing for both photo-inverters is a result of good I-V modulation from HJPT-2. When illuminating HJPT-1 under a range of light intensities, significant difference arises in the photo-inverters' electrical characteristics. In the case of FACs/$C_{16}$-IDTBT, the output voltage signal ($V_{OUT}$) is observed to be only half of the power supply ($V_{DD}$) under low-power illumination. This is attributed to the rather weak photoresponse from HJPT-1, and a pronounced signal inversion was only observed under stronger light intensities (Fig. 6b). The stronger photoresponse of the FACs/$C_8$-BTBT photo-inverter is reflected as the high $V_{OUT}$ that can be achieved at low $V_{IN}$ under the lowest incident light intensity (Fig. 6c). The positions of $V_{IN}$ for the maximum amplification ($A_{i,max}$) remains relatively unchanged under different illumination intensities in the FACs/$C_{16}$-IDTBT photo-inverter (Fig. 6d); whilst for the FACs/$C_8$-BTBT a clear shift towards more negative $V_{IN}$ is observed to achieve $A_{i,max}$ (Fig. 6e). The latter implies a change of device characteristics (i.e., a transistor's threshold voltage) under light illumination. Such a result is in line with the FACs/$C_8$-BTBT HJPT's transfer characteristics (Supplementary Fig. 4). There is no apparent shift observed in the threshold voltage for the FACs/$C_{16}$-IDTBT HJPT because the photogenerated electrons are not 'preserved' in the FACs layer. The latter is supported by our spectroscopic characterisations (Fig. 3) that show the photocarrier lifetime in the FACs is much shorter in the case of FACs/$C_{16}$-IDTBT, indicating a result of nonradiative recombination losses. More importantly, with this unipolar configuration our HJPTs are able to deliver a high $A_{i,\,max}$ of ~15, especially if one considers that this is achieved without any advanced device engineering and only based on the use of a very low-k gate dielectric material[56]. Further plotting $V_{OUT}$ as a function of optical power density under different $V_{IN}$ (Fig. 6f, g for FACs/$C_{16}$-IDTBT and FACs/$C_8$-BTBT, respectively), the FACs/$C_8$-BTBT photo-inverter can operate as an electro-optical switch/sensor[57,58] offering higher tuneability/sensitivity using either electrical or optical signals (or both) as compared to the FACs/$C_{16}$-IDTBT configuration. Importantly, our FACs/$C_8$-BTBT HJPT exhibits excellent bias stress stability (Supplementary Fig. 21), demonstrating the feasibility of our approach for practical applications.

## Discussion

We investigated a conceptual hybrid phototransistor approach by integrating an MHP into a series of OSC transistor channels. The resulting phototransistors are characterised by a straddling-gap type-I heterojunction and exhibit state-of-the-art photogain

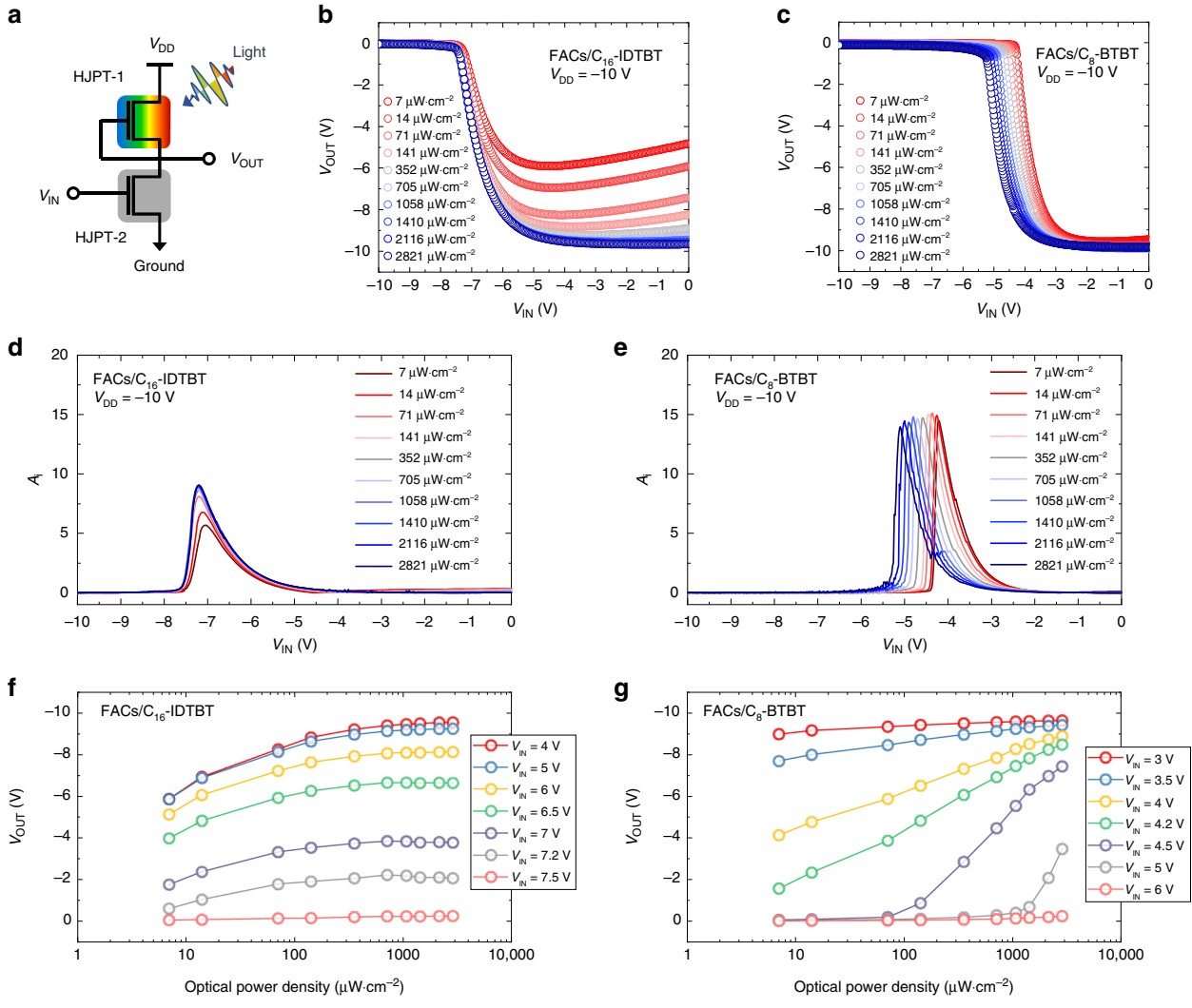

**Fig. 6** Transfer characteristics for HJPT photo-inverters. **a** Schematic of a PMOS-like HJPT photo-inverter to resemble a common-source amplifier for assessing the phototransistor performance metric. $V_{DD}$, $V_{IN}$ and $V_{OUT}$ stand for supply voltage, input voltage signal and output voltage signal, respectively. **b,c**, Transfer characteristics ($V_{OUT}$ vs. $V_{IN}$) measured using a 475-nm LED light source under different incident light intensities for photo-inverters composed of HJPTs using **b** FACs/$C_{16}$-IDTBT and **c** FACs/$C_8$-BTBT. **d, e** Corresponding intrinsic amplification ($A_i$) obtained for **d** FACs/$C_{16}$-IDTBT and **e** FACs/$C_8$-BTBT. **f, g** Effect of incident optical power density on $V_{OUT}$ measured at a fixed voltage supply $V_{DD}$ of −10 V for different $V_{IN}$ biases for the **f** FACs/$C_{16}$-IDTBT photo-inverter and **g** FACs/$C_8$-BTBT photo-inverter

values. Using a variety of characterisations techniques, the high performance can be understood from the efficient photocarrier generation and transport through preferential edge-on molecular orientation at the MHP/OSC heterointerfaces. By tuning such interfaces from the straddling-gap to staggered-gap heterojunction configurations, we demonstrated the ability to tune the optoelectronics of phototransistors in an 'on-demand' manner. Going a step further, we proposed to assess phototransistors using a photo-inverter circuit to evaluate their amplification and essential functionality for the versatile operation in both linear and saturation modes as a transistor. The improved insights into the photocarrier dynamics and its dependence on material properties and operational conditions provide both essential knowledge and thorough understanding for designing and developing perovskite-organic based optoelectronic devices and systems.

## Methods

**Preparation of MHPs and organic semiconductors.** To form the mixed-cation lead mixed anion perovskite precursor solutions, caesium iodide (CsI, Alfa Aesar), formamidinium iodide (FAI, GreatCell Solar), methylammonium iodide (MAI, GreatCell Solar), lead iodide (PbI₂, TCI) and lead bromide (PbBr₂, Alfa Aesar) were

prepared in the way corresponding to the exact stoichiometry for the desired $FA_{0.83}Cs_{0.17}PbI_{2.7}Br_{0.3}$ compositions in a mixed organic solvent of anhydrous N, N-dimethylformamide (DMF, Sigma-Aldrich) and dimethyl sulfoxide (DMSO, Sigma-Aldrich) in a ratio of DMF:DMSO = 4:1. The perovskite precursor concentration used was 0.2 M for making HJPTs and carrying out the UV–vis measurement whilst for the rest of characterisations and measurements in this work the concentration for the perovskite solution was 1.0 M. For organic semiconductors, $C_{16}$-IDTBT was prepared using previously reported procedures in ref. [27] whilst $C_8$-BTBT was purchased from 1-Material and used as received. The organic solutions were prepared by dissolving $C_{16}$-IDTBT and $C_8$-BTBT in tetralin and chlorobenzene, respectively, and stirred for 12 h before use. The blend solutions were prepared by mixing the $C_{16}$-IDTBT and $C_8$-BTBT solutions at desired ratios. The concentration for all organic solutions was 10 mg ml⁻¹.

**Fabrication of heterojunction structures and phototransistors.** TG-BC HJPTs were fabricated on glass substrates. The substrates were cleaned with acetone and isopropanol in a sonication bath for 8 ~ 10 min for each step. Forty nanometres thick Au source and drain electrodes (channel width/length: 1000 μm/30 μm) were evaporated through shadow masks under high vacuum (~10⁻⁶ mbar). The perovskite light-absorbing layer was first deposited using a spin coater with the following processing parameters: starting at 1000 rpm (ramping time of 5 s) for 10 s and then 5000 rpm (ramping time of 5 s from 1000 rpm) for 30 s. Ten seconds before the end of the spinning process, a solvent-quenching method was used by dropping 100 μL toluene onto the perovskite wet films. The organic

semiconductors were deposited from as-prepared solutions via spin coating at 2000 rpm for 30 s and thermally annealed at 100 °C for 5 min. Following semiconductor depositions, a Teflon based material (AF2400, Dupont) dissolved in a Fluorinertt electronic liquid (FC-43, 3 M) at a concentration of 25 mg ml$^{-1}$ was used as the gate dielectric and deposited via spin-cast (500 rpm for 20 s followed by 1000 rpm for 30 s) to produce a ~330 nm thick dielectric layer. The AF2400 gate dielectric was annealed at 50 °C for 30 min. The TG-BC HJPTs were completed by thermal evaporation of 50 nm thick Al gate electrodes through shadow masks under high vacuum (~10$^{-6}$ mbar). The entire transistor fabrication process was carried out in a nitrogen-filled glove box. For perovskite-organic diodes and Mott-Schottky analysis, the samples were using the same cleaning, film-deposition and metal-evaporation processes, except for (i) 20 nm Au bottom electrodes evaporated for perovskite-organic diodes; (ii) ITO-coated glass used for carrying out impedance spectroscopy (IS) in order to perform Mott-Schottky analysis.

**Electrical and optoelectronic characterisations.** Electrical characterisation of transistors and diode was carried out using an Agilent B2902A Precision Source/Measure Unit (SMU) and a homemade probe station in a nitrogen-filled glove box. The SMU was connected to the transistor/diode electrodes via a set of metallic needles held by micropositioners (EB-700, EVERBEING). Characterisation of inverter circuit was carried out using a Keithley 4200 Semiconductor Characterisation System (SCS) in combination with the same probe station. The interconnection of two constituent transistors and the connection to the SCS were achieved via micro-positioning needles and BNC cables.

Optoelectronic characterisation was carried out by measuring the current–voltage characteristics of the fabricated devices in dark environment and under various illumination intensities using light-emitting diodes (LEDs). Specifically, LEDs with electroluminescence emission peaks at 475, 525 and 632 nm were employed. The various illumination intensities were achieved by driving the LEDs with different currents using a Keithley 2400 source-meter. The actual illumination optical powers were calibrated by a Thorlabs 120UV power sensors prior to the measurement.

**Calculation of figures of merit for photodetectors.** Photoresponsivity (R) is an important figure of merit for photodetectors and is defined as the ratio of photo-generated current ($I_{ph}$) to the incident optical power ($P_{opt}$). It can be calculated by the equation:

$$R = \frac{I_{ph}}{P_{opt}} = \frac{I_{illum} - I_{dark}}{E_{opt} \cdot a} \quad (1)$$

where, $I_{illum}$ is the current under illumination, $I_{dark}$ is the current in dark, $E_{opt}$ is the optical power density measured by the power sensors and $a$ is the device area. The device area for transistors is the area enclosed by the channel length and channel width, while that for diodes is defined by the overlap between two electrodes.

Photogain (G) defines the efficiency of optical power converted into measurable electrical current and can be calculated by the equation:

$$G = \frac{I_{ph}}{P_{opt}} \cdot \frac{h\nu}{q} \quad (2)$$

where $h\nu$ is the photon energy. In photodiode an absorbed photon can generate at best one electron-hole pair, corresponding to the maximum obtainable gain of 1. In contrast, the ohmic contact in photoconductor and phototransistor favours charge carriers circulation for several cycles before recombination, hence every single absorbed photon can result in more than one electron detected, i.e., photogain larger than unity is possible. The gain mechanism in such structures can be explained by a trap-assisted model and the corresponding photoconductive gain ($G_{pc}$) can be expressed by[7]

$$G_{pc} = \frac{\tau}{t} \quad (3)$$

where $\tau$ is the photocarrier lifetime and $t$ is the carrier transit time.

Specific detectivity ($D^*$) is another commonly adapted figure of merit to quantify the sensitivity of a photodetector with respect to noise and enable direct device comparison with different areas and geometries. The equation of $D^*$ is given as

$$D^* = \frac{\sqrt{a \cdot f_B}}{NEP} = \frac{\sqrt{a \cdot f_B} \cdot R}{S_{noise}} \quad (4)$$

where $a$ is the device area, $f_B$ is the bandwidth of the device, NEP is the noise-equivalent power, $R$ is the responsivity and $S_{noise}$ is the noise spectral density. The noise spectral density in this study is acquired using a Fourier transform-based method (see Supplementary Note 5).

**UV–vis spectroscopy.** The UV–vis transmission measurements were performed using a Shimadzu UV-2600 UV–vis spectrophotometer. The samples were prepared on quartz substrates using the same deposition parameters described earlier.

**Atomic force microscopy.** Atomic force microscopy study was carried out in tapping mode using an Agilent 5500 atomic force microscope in ambient atmosphere. The approximate resonance frequency of the cantilever was 280 kHz and force constant was ≈60 Nm$^{-1}$.

**Grazing-incidence wide-angle X-ray scattering.** Grazing-incidence wide-angle X-ray scattering (GIWAXS) measurements were conducted on CMS beamline at NSLS II, Brookhaven National Lab. The x-ray with the energy of 13.5 keV shone on the as-cast films at various incident angles, including 0.05° and 0.1°. The high incident angle of 0.1° allows the X-ray to penetrate through the top organic layer into the bottom perovskites layer. A Photon Science CCD detector was placed 227 mm away from the sample and recorded for 10 s for each film. The data analysis was performed by SciAnalysis programme.

**Photoluminescence spectroscopy.** Steady-state photoluminescence spectra were recorded using an excitation wavelength of 510 nm and slit widths of 5 mm on a commercial spectrofluorometer (Horiba, Fluorolog). Time-resolved PL measurements were acquired using a time-correlated single photon counting (TCSPC) setup (FluoTime 300 PicoQuant GmbH). Film samples were photoexcited using a 507 nm laser head (LDH-P-C-510, PicoQuant GmbH) pulsed at 0.2 MHz.

**Transient photoconductivity.** A schematic drawing for the TPC measurement is shown in Supplementary Fig. 13. A Nd:YAG pulsed laser excitation source pumped at 10 Hz with FWHM = 3.74 ns was set to 470 nm with a range of fluences in order to vary the charge carrier densities of FACs. A bias of 24 V is applied on one of the in-plane electrodes. A variable resistor is connected in series with the samples. The voltage drop on this resistor was monitored through an oscilloscope with a high internal resistance (1 MΩ) connected in parallel to determine the change of the potential across the two in-plane Au electrodes. TPC was calculated using the following equation, $\sigma_{TPC} = \frac{V_r}{R_r \times (V_{appl} - V_r)} \times \frac{l}{w \times t}$, where $R_r$ is resistance for the variable resistor, $V_r$ is the potential drop measured across the resistor, $V_{appl}$ is applied voltage, $l$ is channel length, $w$ is channel width, and $t$ is film thickness. For sample preparation, the as-deposited films were scribed to have a 5-mm channel width and coated with a layer of 200-nm poly(methyl methacrylate) (PMMA) whilst the channel length is defined by two thermally-evaporated Au electrodes with a distance of 4 mm.

**Time-resolved microwave conductivity.** A diagrammatic illustration of the TRMC system used in this study is shown in Supplementary Fig. 10. A microwave-frequency oscillatory electric signal is generated using a Sivers IMA VO4280X/00 voltage-controlled oscillator (VCO). The signal has an approximate power of 16 dBm and a tuneable frequency between 8 and 15 GHz. The oscillatory signal is incident on an antenna inside a WR90 copper-alloy waveguide. The microwaves emitted from the antenna pass through an isolator and an attenuator before they are incident on a circulator (Microwave Communication Laboratory Inc. CSW-3). The circulator acts as unidirectional device in which signals entering from port 1 exit through port 2 and signals entering from port 2 exit through port 3. The incident microwaves pass through a fixed iris (6.35 mm diameter) into a sample cavity. The cavity supports a TE$_{102}$ mode formed by a short section of WR90 waveguide and an ITO-coated glass window that allows optical access to the sample. The sample is mounted inside the cavity at a maximum of the electric-field component of the standing microwaves, using a PLA sample holder.

Microwaves reflected from the cavity are then incident on port 2 of the circulator, exiting through port 3, directed through an isolator, and directed to a zero-bias Schottky diode detector (Fairview Microwave SMD0218). The detector outputs a voltage linearly-proportional to the amplitude of the incident microwaves. The signal is amplified by a Femto HAS-X-1-40 high-speed amplifier (gain = ×100). The amplified signal is detected by a Textronix TDS 3032 C digital oscilloscope. A Continuum Minilite II pulsed Nd:YAG laser is used to illuminate the sample. The laser pulse has a wavelength of 532 nm, a full-width at half-maxima of ~5 nm and a maximum fluence incident on the sample of ~10$^{15}$ cm$^{-2}$. An external trigger link is employed to trigger the oscilloscope ~50 ns before the laser fires. The area exposed to the incident optical pulse is approximately 25% of the cross-sectional area of the cavity. Changes in the detector voltage under illumination can then be used to extract the relevant TRMC parameters: $\phi\Sigma\mu_{TRMC}$, as described in Supplementary Note 3.

**Scanning electron microscopy.** A Hitachi S-4300 scanning electron microscope was used to acquire cross-sectional images of target samples.

**Impedance spectroscopy.** The IS technique was carried out using an Autolab PGSTAT302N potentiostat with a FRA32M module to perform the Mott-Schottky analysis. In the IS measurement, the AC perturbation was set to 10 mV. The illumination was supplied by a LED light source with a flux of ~12 lumens. The room conditions were of ~20–30% humidity and 295–300 K temperature.

**High-angle annular dark-field scanning transmission electron microscopy**. The sample for cross-sectional HAADF-STEM was prepared by a focused ion beam (FIB) technique using a Helios G4 UX (FEI). About 50 nm of a protective aluminium layer was deposited by a thermal evaporator (Angstrom) before performing the cross-sectional sample preparation by FIB. The HAADF-STEM image of a bilayer sample was investigated by a Titan Themis Z (FEI) transmission electron microscope operated at an acceleration voltage of 300 kV.

**Dynamic secondary ion mass spectrometry**. Depth-profiling experiments were performed on a DSIMS instrument from Hiden analytical company (Warrington-UK) operated under ultra-high vacuum conditions, typically $10^{-9}$ torr. The FACs/ blend 1:1 bilayer stack was deposit on a silicon substrate. To avoid the edge effect, the data were recorded from a small area (typically $75 \times 75$ μm) located in the middle of the sputtered region (typically $750 \times 750$ μm) using an adequate electronic gating.

## Data availability
The data that support the plots within this paper and other finding of this study are available from the corresponding author upon reasonable request.

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

## Acknowledgements

This project was funded by EPSRC, Engineering and Physical Sciences Research Council grants, EP/M005143/1 and EP/P006329/1. T.D.A. acknowledges the King Abdullah University of Science and Technology (KAUST) for the financial support. This work used CMS beamline of the National Synchrotron Light Source II, a U.S. Department of Energy (DOE) Office of Science User Facility operated for the DOE Office of Science by Brookhaven National Laboratory under Contract No. DE-SC0012704.

## Author contributions

Y.-H.L. and H.J.S. conceived the concept of the project, and H.J.S. guided and supervised the project. Y.-H.L. and W.H. contributed equally to the fabrication of the transistors, diodes, circuits and thin-film samples, and performed the electrical and optoelectronic measurements. Y.-H.L. and W.H. analysed all the device data. W.H. carried out AFM and analysed the data with Y.-H.L. W.H. analysed the optoelectronic properties of the HJPTs and provided insights into photodetectors. Y.-H.L. prepared the samples for spectroscopic analysis and carried out capacitance-voltage characterisations. P.P. provided insights into photocarrier dynamics, photodetection mechanisms and assisted in the Mott-Schottky analysis. J.C.L. carried out TPC and analysed the data. R.L. performed GIWAXS and analysed the data. N.S. carried out PL and SEM. J.P. supported the preparation of organic materials and transistors. M.J.H. and J.G.L. carried out TRMC and analysed the data. Z.F. and M.H. synthesised the polymer material. C.M. prepared the samples for HAADF-STEM and DSIMS. N. Wei performed the FIB preparation and HAADF-STEM characterisation. N. Wehbe performed the DSIMS measurement. T.D.A. provided insights into the material, electrical and optoelectronic characterisations and guidance. Y.-H.L. wrote the first draft of the paper. All authors discussed the results and contributed to the writing of the paper.

## Competing interests

The authors declare no competing interests.
