## [Peer Review File · Nature Communications]

Reviewers' comments:

Reviewer #1 (Remarks to the Author):

The authors presented a high-performance MHP/organic semiconductor layered phototransistor with type-I heterojunctions. The authors used a variety of spectroscopic analysis techniques to study the nature of charge-transfer between MHP and OSC. They carried out Mott-Schottky analysis to elucidate the underlying mechanism of efficient photocarrier circulation and high photoresponsivity (also high photogain). And highly efficient photocarrier transfer was found to transport through preferential edge-on molecular orientation at the MHP/OSC heterointerfaces. In addition, they also evaluated the photodetection performance and amplification of phototransistors by using the photo-inverter circuit.

The methodology is based on the introduction of "straddling-gap" type-I heterojunctions, which is useful for optoelectronic devices requiring a quantum well structure, such as light-emitting diodes (LEDs). The presented results are at first sight counterintuitive: among the two types of heterojunctions, type-II (staggered gap) heterojunction was commonly considered as the optimal candidate for photodetectors due to its ability to enhance the exciton dissociation and charge transfer process. I am impressed by the employment of type-I heterojunction in phototransistor, as type-I heterojunction cannot facilitate the charge transfer process so that the photogenerated holes in FACS cannot inject to the OSC channel, and the PL spectral indeed exhibits inefficient charge transfer between FACS and OSC, but high photogain and high photoresponsivity were obtained in the type-I heterojunction based phototransistors. The employment of type-I heterojunction in phototransistor is new and the results are very interesting. A more in-depth analysis and discussion are needed before its publication on Nature Communications.

1) The most appealing point of this manuscript is the employment of type-I heterojunction, which leads to the pronounced photoresponse at the off-state. The authors should leverage this point, the most important novelty of this work, in the introduction part, rather than just describe its impact on the development of perovskite phototransistor. I believe this work might have a broader and more general impact if the motivation of using type-I heterojunction in the phototransistors is explained more clearly.

2) The device structure used in this work does not appear to be optimal because of the gap between S/D electrodes and the conduction channel (OSC layer). In this type of structure, the injected carriers have to traverse through the FACS layer before they travel to the OSC channel, which de affect the carrier circulation and reduce the consequent photocurrent (photoresponsivity and photogain). The authors should explain this.

3) For the AFM images in Figure 1d, it seems that the FACS layer was not completely covered by the upper organic semiconductor layer (C8-BTBT or blend 1:1).

4) The perovskite/organic semiconductor phototransistors and organic/organic phototransistors based on type-II heterojunction have been reported. (e.g., "A novel hybrid-layered organic phototransistor enables efficient intermolecular charge transfer and carrier transport for ultrasensitive photodetection" *Adv. Mater.* 31, 1900763, 2019; "High-performance inorganic perovskite quantum dot-organic semiconductor hybrid phototransistors" *Adv. Mater.* 29, 1704062, 2017). The decent photodetection in terms of high photoresponsivity and high EQE & photogain have been obtained based on such type-II heterojunction hybrid structures. The authors should provide an in-depth discussion to compare the device using type-I heterojunction with above published works and illuminate the advantages and necessities of using type-I heterojunction in phototransistors.

5) The significant positive shift of the threshold voltage and high photoresponse have already been reported in phototransistors with various architectures. Generally speaking, the injected photocarriers from the light absorber would transport laterally through the channel and contribute to the photocurrent. In principle, the trapped photoelectrons in the FACS layer should lead to the positive shift of the threshold voltage. However, in Figure 2a, the transfer curves under light and in the dark appear to merge together for the FACS/C16-IDTBT devices, which implies the very limited photoresponse. Now that the relatively good responsivity and efficient charge transfer are

observed in the FACs/C16-IDTBT type-II heterojunctions (gate-voltage dependent R & PL spectral), why there is no positive shift of the threshold voltage and high photocurrent? The reason should be elaborated.

6) When the authors tried to explain the role of type-I heterojunction in the phototransistor, different techniques, including PL, TRMC, Mott-Schottky analysis were performed. However, the techniques used by the authors are mainly based on the measurements of thin films or their stacks. The measurements for the full devices, e.g., the mobility of different devices, should also be provided to exhibit the impact of type-I heterojunction.

7) A TEM or SEM picture showing the cross-section of the devices should be provided to clearly demonstrate the device interfaces.

8) The unit for y-axis should be added in figure 2a.

9) There are some typos such as "C16-BTBT" (page 4 and Figure 3b,c), etc.

10) The authors should carefully check the reference format, e.g., ref 19 & 25.

Reviewer #2 (Remarks to the Author):

Review comments:

The manuscript entitled "Deciphering photocarrier dynamics for tuneable, high-performance perovskite-organic semiconductor heterojunction phototransistors" has reported a conceptual hybrid heterostructure involving metal-halide perovskite MHP (hereafter FACs) and organic small molecular film (OSCs), and found great different optoelectric response behaviors under "on-state" and "off-state", and these optoelectric response can also be gate-tuned with various combination between FACs and blend content of C8-BTBT: C16-IDTBT. More importantly, it is very interesting to build up a photo-inverter by a strategy applying for unipolar phototransistors made by these FACs/OSCs heterojunctions. The key point of the research is to try to decipher the dynamics behind photocarriers' charge transfer process, as well as photoresponse mechanisms in the heterostructures. However, present version of the manuscript contains too much data and puzzled explanations which make the main point unclear. Some of my suggestions and questions are listed below in order for providing to authors for their improvement.

1). For all the data, the most important results perhaps are those experimental one illustrated in Figure 2(a) and schematic draw in Figure 5(a-c). However, by employing the information provided from the figure 5, I was not able to decipher the difference shown in Figure 2(a), e.g. for the large absolute value of negative bias (on-state), the drain current I_D is smaller for FACs/C8-BTBT than FACs/C16-IDTBT, in contrast the photoresponse is much larger for FACs/C8-BTBT than FACs/C16-IDTBT. For the low bias voltage (off-state), the same trend of photoresponse can be uncovered as those found under "on-state".

To provide a reasonable explanation, the authors resorted to type-I and type-II band alignment as well as the specific structural features shown in small molecular arrangement (face-on and edge-on, ect.). However, the authors did not persuade me to believe the scenario appeared in Figure 5.

2). As mentioned above, we do not know how the carrier density distributes across the interface near the heterojunction, as the authors did not even provide the thickness of FACs and OSC films. I just wonder how dependence on the heterojunction interface between the field-effect character and the thickness of OSC film?

3). Because the picture for the illustrative photocarrier charge transfer is unclear, the vast group of characterization targeting for the photoelectric technique is logically unnecessary. Moreover, I strongly suggest not use so much supplementary notes and figures, as one paper however important can not contain so huge amount of data.

4). There were still some spelling errors or miss in the context, such as:

Page 11/bottom line, Fig. 6b and 6c should be FACs/C16-IDTBT and FACs/C8-BTBT, respectively. Figure 3b of the inset should be FACs/C16-IDTBT. etc.

Reviewer #3 (Remarks to the Author):

This manuscript demonstrated a hybrid heterojunction phototransistor by combining a metal halide perovskites (MHP) beneath a stack of organic semiconductor (OSC) transistor channels. By utilizing two different OSCs (C8-BTBT and C16-IDTBT) with different band structures, either "straddling-gap" or "staggered-gap" heterojunctions has been achieved. The former phototransistor exhibited state-of-the-art photogain value because of preferential edge-on molecular orientation resulted highly efficient photocarrier cycling. The optoelectronic property of phototransistors could also be tuned by changing the device between those two different types. The device concept is of novelty and this manuscript has shown systematic experimental investigation and theoretical analysis on the device with demonstration of performance enhancement. However this reviewer believes the following points should be well addressed before acceptance can be recommended:

(1) Can the authors provide more data on pristine perovskite based phototransistors to show how the polymers (C8-BTBT and C16-IDTBT) improve the photodetection performance?

(2) The authors claim that for C8-BTBT, the carriers transport through preferential edge-on molecular orientation at the MHP/OSC hetero-interfaces, therefore, leading to improved photodetection performance. The surface of perovskite may affect the device performance. Can the authors provide a clear image to show the perovskite morphology?

(3) In the introduction, the authors claim that addition conductive channel is inserted to increase the photoconductive gain. However, the TRMC results showed that the perovskite has higher mobility than the polymers. Can the authors explain more why they choose these two polymers to modify perovskite?

(4) By characterizing the PL and TRPL, the authors claimed that the OSC material in the blend in direct contact with FACs is mostly C16-IDTBT, which was also in line with the reported vertical phase separation. Can the author provide more straightforward evidences, such as SIMS (a surface analytical technique), to show the vertical material distribution?

(5) The authors attributed the reason for the shorter PL decay and quenched PL signal of FACs/C16-IDTBT to more effective transfer of photocarriers from FACs to OSC, which is, however, in conflict with the band diagram shown in Figure 1c. More specifically, the higher HOMO level of C8- BTBT in theory should make the hole transfer easier than that C16-IDTBT. More discussion is need.

(6) For the control device, except for the blend OSC material, can the authors provide one more device with layer by layer deposited C16-IDTBT and C8-BTBT, with either the former or the latter in contact with perovskite layer? Will it show similar performance with the blend one?

(7) Some minor errors: the full name of OSC is missing, light intensity unit in Figure 6b is incorrect.

Reviewers' comments:

=====

Reviewer #1 (Remarks to the Author):

The authors presented a high-performance MHP/organic semiconductor layered phototransistor with type-I heterojunctions. The authors used a variety of spectroscopic analysis techniques to study the nature of charge-transfer between MHP and OSC. They carried out Mott-Schottky analysis to elucidate the underlying mechanism of efficient photocarrier circulation and high photoresponsivity (also high photogain). And highly efficient photocarrier transfer was found to transport through preferential edge-on molecular orientation at the MHP/OSC heterointerfaces. In addition, they also evaluated the photodetection performance and amplification of phototransistors by using the photo-invertor circuit.

The methodology is based on the introduction of “straddling-gap” type-I heterojunctions, which is useful for optoelectronic devices requiring a quantum well structure, such as light-emitting diodes (LEDs). The presented results are at first sight counterintuitive: among the two types of heterojunctions, type-II (staggered gap) heterojunction was commonly considered as the optimal candidate for photodetectors due to its ability to enhance the exciton dissociation and charge transfer process. I am impressed by the employment of type-I heterojunction in phototransistor, as type-I heterojunction cannot facilitate the charge transfer process so that the photogenerated holes in FACs cannot inject to the OSC channel, and the PL spectral indeed exhibits inefficient charge transfer between FACs and OSC, but high photogain and high photoresponsivity were obtained in the type-I heterojunction based phototransistors. The employment of type-I heterojunction in phototransistor is new and the results are very interesting. A more in-depth analysis and discussion are needed before its publication on Nature Communications.

AUTHORS RESPONSE: We really appreciate that the reviewer has shown great interest in our approach. Not only we have addressed the reviewer’s comments point by point in our responses below and revised our manuscript accordingly, but we’ve also revised many parts of our manuscript in order to provide more in-depth analysis for the mechanism behind our approach. We sincerely hope the reviewer finds our revision satisfactory.

REVIEWER: 1) The most appealing point of this manuscript is the employment of type-I heterojunction, which leads to the pronounced photoresponse at the off-state. The authors should leverage this point, the most important novelty of this work, in the introduction part, rather than just describe its impact on the development of perovskite phototransistor. I believe this work might have a broader and more general impact if the motivation of using type-I heterojunction in the phototransistors is explained more clearly.

AUTHORS RESPONSE: We thank the reviewer for the recognition of the importance of this work. We have revised the introduction in our revised manuscript with a clearer statement to describe our motivation of the employment of type-I heterojunction. We hope the reviewer finds the revision satisfactory.

REVIEWER: 2) The device structure used in this work does not appear to be optimal because of the gap between S/D electrodes and the conduction channel (OSC layer). In this type of structure,

the injected carriers have to traverse through the FACs layer before they travel to the OSC channel, which de affect the carrier circulation and reduce the consequent photocurrent (photoresponsivity and photogain). The authors should explain this.

AUTHORS RESPONSE: We thank the reviewer for raising this important comment. We agree with the reviewer on the point that the sandwich structure might result in poor charge injection under the circumstance that there is only weak gate-induced electric field (e.g. a transistor's off-state), as compared to the case where the source and drain (S/D) electrodes are in direct contact with the transistor channel. For the current submitted work, we in fact had carried out the other type of device configurations as shown in **Fig. R1a** where the S/D electrodes are deposited on top of the FACs perovskite light absorber and in contact with the organic semiconductor (OSC – blend 1:3, i.e. C₈-BTBT : C₁₆-IDTBT = 1:3) transistor channel. We did observe a higher increase in the photocurrent at the off-state (**Fig. R1a**), and the drain current level is close to the FACs perovskite-only phototransistor (**Fig. R2a**). As such, the high photocurrent in the off-state could be attributed to the charge transport in the perovskite layer.

Fig. R1. | Impact of device architectures on photoresponse. Transfer characteristics for the HJPTs in a configuration with (a) the S/D electrodes in direct contact with the OSC blend 1:3 (i.e. C₈-BTBT : C₁₆-IDTBT = 1:3) layer; and (b) the FACs perovskite light absorber sandwiched in-between the S/D electrodes and the OSC layer. The illumination was performed using a green light ($\lambda_{\text{peak}} = 525 \text{ nm}$) of $705 \mu\text{W cm}^{-2}$.

Fig. R2. | Optoelectronic characterisation of HJPTs. Transfer characteristics of phototransistors with (a) FACs only and (b) FACs/C₈-BTBT heterojunction measured in the dark and under green illumination ($\lambda_{\text{peak}} = 525 \text{ nm}$) of $705 \mu\text{W cm}^{-2}$. Both devices were operated at a constant $V_D = -40 \text{ V}$. Phototransistors with FACs-only showed low I_D of $\sim 10^{-10} \text{ A}$ in the dark and enhanced I_D of $> 10^{-9} \text{ A}$ under illumination. These results suggest that the FACs film is not capable of transporting sufficient channel current (i.e., I_D) to form a transistor. In contrast, the FACs/C₈-BTBT heterojunction phototransistor exhibited i) excellent current-voltage modulation by varying gate bias voltages and ii) a significant increase in I_D at the off-state to $> 10^{-7} \text{ A}$ under illumination. As such, combining FACs with OSC (i.e., C₈-BTBT here) is the key to form a fully functioning phototransistor with excellent photoresponse.

However, the photoresponse nearly vanishes when the phototransistor is operated beyond the turn-on voltage (**Fig. R1a**), similar to what has been reported in the literature (*Adv. Electron. Mater.* 2017, **3**, 1600325). This observation is attributed to the following reason: the S/D electrodes are located above the light absorber (i.e. FACs perovskite), and hence there is no effective gate-induced electric field to transfer photocarriers from the light absorber to the transistor channel. Although one might still argue that there is an in-plane electric field established by the potential difference between the S/D electrodes, however, this in-plane field is much weaker compared to the out-of-plane electric field induced by the gate bias, simply because the S/D electrodes are several micrometres apart. In addition, the relative position of the perovskite layer in the phototransistor (**Fig. R1a**) resembles a planar top-gate top-contact (TG-TC) device architecture that could not benefit from the stronger field effect induced by the gate bias. As such, the photocarriers generated from the light absorber could not be efficiently transferred to the OSC transistor channel as the photocarriers sit outside the effective region that is under the influence of the strong gate-induced field effect, i.e. the region in-between the S/D electrodes and the gate dielectric. The photocarriers accumulated in the light absorber could cause Coulombic scattering (*Adv. Mater.* 2016, **28**, 3952–3959; *Adv. Mater.* 2015, **27**, 176–180) and eventually reduce the photocurrent at the on-state (**Fig. R1a**, also see *Adv. Electron. Mater.* 2017, **3**, 1600325).

On the other hand, **Fig. R1b** shows the corresponding top-gate bottom-contact (TG-BC) device architecture that sandwiches both the FACs and OSC layers using the same material composition as the device studied in **Fig. R1a**. With such a TG-BC configuration, the FACs layer is in the region under the influence of a strong out-of-plane gate-induced electric field, which helps a more efficient photocarrier transfer from the perovskite light absorber to the OSC transistor channel. The transferred photocarriers (i.e. photoholes) increase the carrier density within the transistor channel and contribute to the charge carrier circulation, leading to an increase in the transistor drain current (*Adv. Mater.* 2013, **25**, 4267–4295). Meanwhile, the photoelectrons left inside the perovskite layer will accumulate around the source electrode under the applied gate field, lowering the injection potential barrier between the electrode and the transistor channel. The change in the injection barrier under illumination causes a threshold voltage shift in a phototransistor (*Adv. Mater.* 2013, **25**, 4267–4295). The latter is the so-called photovoltaic effect, which is desired from the viewpoint of phototransistor operation as it creates an optimal operational regime near the transistor subthreshold region where dark current and photogain are optimised simultaneously (*ACS Photonics.* 2016, **3**, 2197–2210). By comparing the two

configurations shown in **Fig R1**, one can easily find that the configuration sandwiching both the light absorber and transistor channel (**Fig. R1b**) exhibits an enhanced photovoltaic effect (i.e. larger threshold voltage shift), which can be explained by the fact that this configuration with the light absorber sandwiched by the S/D electrodes and the OSC channel allows a more sensitive injection barrier variation by light. Having said that, we have provided more in-depth discussion on the case of the FACs/C₁₆-IDTBT HJPT in the response to the reviewer's comment #5 regarding the low photocurrent and nearly negligible threshold voltage shift, even using the device that has been configured in the same way as shown in **Fig R1b**.

To this end, we would like to point out there is an apparent trade-off of between photoresponse and charge injection from S/D electrodes for different transistor operation regimes. Using our straddling-gap HJPT (**Fig. R2b**), we have demonstrated that our approach can effectively mitigate the limitation caused by device architectures and enhance the photoresponse across the entire operation regime. We hope our response has provided more insight here.

REVIEWER: 3) For the AFM images in Figure 1d, it seems that the FACs layer was not completely covered by the upper organic semiconductor layer (C8-BTBT or blend 1:1).

AUTHORS RESPONSE: We thank the reviewer to point out this concern. For the deposition of small-molecule rich organic solutions (i.e. C₈-BTBT or even blend 1:1), there is a great opportunity that the films could form material-rich/material-deficient regions due to the strong crystallisation property of small molecules, hence resulting in a discontinuous layer (*Org. Electron.* 2016, **36**, 73-81). One way to overcome this disadvantage is to adopt small molecule/polymer blends (*Adv. Mater.* 2016, **28**, 7791–7798; *Nat. Commun.* 2014, **5**, 3005) to form a continuous organic layer, with the help from the amorphous property of polymers. In our HJPT, the texture of the FACs perovskite layer could also post a challenge to the formation of a continuous layer in the subsequent deposition (i.e. organic semiconductors). Nevertheless, in this work we have achieved a hole field-effect mobility of 0.33 and 0.42 cm²·V⁻¹·s⁻¹ for HJPTs based on C₈-BTBT and blend 1:1, respectively. These performance parameters are two orders of magnitude higher than the reported values in the literature (e.g. the results obtained from conventional spin-cast techniques, *Org. Electron.* 2016, **36**, 73-81). Also, we would like to point out there have been several reports using modified deposition techniques that have been demonstrated to largely enhance field-effect mobilities in organic thin-film transistors (e.g. *Nature* 2011, **480**, 504–508; *Nat. Mater.* 2013, **12**, 665–671; *Nat. Commun.* 2014, **5**, 3005). On the other hand, the film stresses in the FACs perovskite could in principle be reduced by different processing protocols (*ACS Energy Lett.* 2018, **3**, 1225–1232). As such, we anticipate our HJPT device performance can be further improved by adopting different deposition techniques and processing protocols. However, the current work focuses on the concept of how to effectively extract photocarriers from the perovskite-based light absorber by bandgap engineering using different types of heterostructures. Therefore, the optimisation of the processing engineering will be beyond the scope of the current submission. We really appreciate the reviewer's observation and hope the reviewer finds this response satisfactory.

REVIEWER: 4) The perovskite/organic semiconductor phototransistors and organic/organic phototransistors based on type-II heterojunction have been reported. (e.g., “A novel hybrid-layered organic phototransistor enables efficient intermolecular charge transfer and carrier transport for ultrasensitive photodetection” *Adv. Mater.* 31, 1900763, 2019; “High-performance inorganic

perovskite quantum dot-organic semiconductor hybrid phototransistors” *Adv. Mater.* 29, 1704062, 2017). The decent photodetection in terms of high photoresponsivity and high EQE & photogain have been obtained based on such type-II heterojunction hybrid structures. The authors should provide an in-depth discussion to compare the device using type-I heterojunction with above published works and illuminate the advantages and necessities of using type-I heterojunction in phototransistors.

AUTHORS RESPONSE: We thank the reviewer to point out these two examples from the literature using the concept of type-II heterojunction in the design of phototransistors. In the first example (“A novel hybrid-layered organic phototransistor enables efficient intermolecular charge transfer and carrier transport for ultrasensitive photodetection” published in *Adv. Mater.* 2019, **31**, 1900763), a C₈-BTBT : PC₆₁BM bulk heterojunction is used as the photoactive layer (i.e. the light absorber), where bound excitons with large binding energies are created under light illumination. The C₈-BTBT : PC₆₁BM type-II bulk heterojunction is therefore used to enhance the exciton dissociation by the energy offset between the donors and the acceptors. What is the most interesting part in this report is that the authors managed to implement a layer of MoO₃, a well-known hole-injection (electron-blocking) interlayer in the applications of organic photovoltaics and organic light-emitting diodes, not only in-between the transistor channel and the S/D contacts but also in-between the light absorber and the transistor channel as a multi-functional interlayer for the following three main functionalities (adopted from *Adv. Mater.* 2019, **31**, 1900763): “1) decreasing the contact resistance at the metal/organic interface and improving the hole mobility, 2) facilitating hole injection while blocking electrons from the BHJ to the channel and further suppressing exciton recombination, and 3) preventing the underneath channel layer from degradation during the solution-based coating of the photoactive layer.” We notice that the transistor channel is composed of C₈-BTBT. Therefore, this C₈-BTBT transistor channel forms a “homojunction” like HOMO level with that of the C₈-BTBT : PC₆₁BM bulk heterojunction light absorber for photohole transport, rather than a heterojunction. However, in our HJPT approach we purposefully form a heterojunction between the light absorber and the transistor channel, which is completely different from this example (*Adv. Mater.* 2019, **31**, 1900763).

One of the intriguing parts of this example is that the insertion of a layer of MoO₃ could effectively block the transport of photoelectrons, hence minimising the chances of the recombination that could happen between photoelectrons and photoholes as pointed out by the authors of *Adv. Mater.* 2019, **31**, 1900763. If one considers this MoO₃ interlayer that sits between the transistor channel and light absorber, the photoholes generated in the bulk heterojunction actually undergo i) type-II heterojunction (the interface between the C₈-BTBT : PC₆₁BM bulk heterojunction and the MoO₃ interlayer), then ii) type-I heterojunction (the interface between the MoO₃ interlayer and the C₈-BTBT transistor channel). Due to the ultra-thin thickness of the MoO₃ interlayer and the similarity of the energetic levels between the VB of MoO₃ and the HOMO of C₈-BTBT, the photoholes are most likely tunnelling through this interlayer. The most critical part of this work, we believe, is the use of C₈-BTBT for both the light absorber and the transistor channel as a continuous layer of C₈-BTBT serves as an excellent field-effect channel. On the other hand, in the case of using perovskite as the light absorber the relatively low exciton binding energy in perovskites enables free charges to be generated following light absorption (*Nat. Nanotechnol.* 2015, **10**, 391–402). The challenge for the perovskite-based devices to achieve high photoresponsivity and photogain will be how to find an effective way to extract photocarriers. We further detailed relevant discussion regarding nonradiative losses in our response to the reviewer’s next comment, which could serve a complimentary explanation to the current comment. In short,

we found that engineering a type-I heterojunction between FACs/OSCs using C₈-BTBT helps charge transport along the transistor channel direction as C₈-BTBT molecules form an edge-on orientation. Moreover, long-lived photocarriers are observed in FACs/C₈-BTBT as evidently seen in the spectroscopic results using several different techniques, i.e. PL, TRMC and TPC (**Fig. 3**).

As what has been briefly described in the supplementary information of our manuscript, we found that most of the phototransistor works in the literature employ a type-II heterojunction configuration for the light absorber and the transistor channel. Although this method has been proven to be intuitive and effective with various architectures, photodetection performance of phototransistors based on this concept strongly depends on the field-effect mobility of the channel materials (**Supplementary Note 1**). The hybrid perovskite phototransistors with high performance are often based on materials that show promising field-effect mobilities, such as graphene (*ACS Appl. Mater. Interfaces* 2017, **9**, 1569–1576) and carbon nanotube (*Nanoscale*. 2016, **8**, 4888–4893). The implementation of high mobility materials increases the current level not only under light illumination but also in the dark, resulting in a trade-off between photogain and dark current densities (*Nature*. 2017, **542**, 324–327).

As for the second example pointed out by the reviewer (“*High-performance inorganic perovskite quantum dot-organic semiconductor hybrid phototransistors*” *Adv. Mater.* 2017, **29**, 1704062), this work employs a different active layer configuration (i.e. gate-dielectric/light-absorber/organic-semiconductor) than ours (gate-dielectric/organic-semiconductor/light absorber). In this example, the incident light needs to travel through the organic transistor channel to reach the light absorber, which could lead to a loss of light. Meanwhile, the phototransistor performance would be degraded as the channel (i.e. organic semiconductor) could not be effectively subjected to the gate-induced field effect due to the presence of a layer of perovskite quantum dots. The latter is evidently seen in Fig. 2d in the main text of *Adv. Mater.* 2017, **29**, 1704062 and in Fig. S2 in the supporting information of *Adv. Mater.* 2017, **29**, 1704062 – i.e. using the organic material alone shows a higher drain current and a turn-on voltage close to $V_G = 0$ V. In comparison, we believe our HJPT approach has higher versatility as there is still plenty of room for improving our HJPT’s performance through various deposition techniques, as seen in our response to the reviewer’s previous comment.

On the other hand, we would like to emphasise that this example (*Adv. Mater.* 2017, **29**, 1704062) employing CsPbBr₃ quantum dots as the light absorber, not a perovskite thin film. This difference in materials could make a huge difference as there exist some forms of organic ligands on the surfaces of the quantum dots (see Experimental Section in *Adv. Mater.* 2017, **29**, 1704062). If we take this into account, the ligands could act as a barrier and in fact, result in a type-I heterojunction configuration between the light absorber and transistor channel. Since more details regarding the energetic level for the ligand are not revealed in this specific report (*Adv. Mater.* 2017, **29**, 1704062), it is not possible for us to very thoroughly discuss the matter regarding the advantages for type-I and type-II heterojunction configurations in order to compare our work with *Adv. Mater.* 2017, **29**, 1704062. Nevertheless, we note that due to their device configuration (i.e. gate-dielectric/light-absorber) the applied gate field can ‘pull’ the channel carriers (i.e. holes) close to where the perovskite light absorber is. Together with the barrier formed by the ligand, nonradiative recombination losses at the interface between the light absorber and transistor channel might be minimised. As such, regardless of what type of heterojunctions that forms at the interface between the light absorber and transistor channel [despite the latter, we would like to emphasise again that the ligand(s) used for the quantum dots could result in an energy barrier, so there should not exist a simple type-II heterojunction relationship between the light absorber and transistor

channel in the case of *Adv. Mater.* 2017, **29**, 1704062], the effective transistor channel still forms on the side of the organic material as the CsPbBr₃ quantum-dot only transistor does not exhibit a sufficient drain current level to have influence on the transistor's current-voltage behaviours (see Fig. S3 in *Adv. Mater.* 2017, **29**, 1704062).

On a separate note, the active materials used in our HJPTs are fully processed from solutions whilst the approaches shown in *Adv. Mater.* 2017, **29**, 1704062 (MoO₃ deposited by thermal evaporation) & *Adv. Mater.* 2017, **29**, 1704062 (organic DNTT deposited by vacuum evaporation) both require the use of vacuum-based deposition techniques to complete the active layers. The ability to implement solution-only processes could enable future applications in large-area flexible optoelectronics (*Adv. Mater.* 2017, **29**, 1702838.). To this end, we would like to thank the reviewer to point out these two important reports for us. We have added relevant discussion regarding their content in our revised manuscript. We hope the reviewer finds our response satisfactory.

REVIEWER: 5) The significant positive shift of the threshold voltage and high photoresponse have already been reported in phototransistors with various architectures. Generally speaking, the injected photocarriers from the light absorber would transport laterally through the channel and contribute to the photocurrent. In principle, the trapped photoelectrons in the FACs layer should lead to the positive shift of the threshold voltage. However, in Figure 2a, the transfer curves under light and in the dark appear to merge together for the FACs/C16-IDTBT devices, which implies the very limited photoresponse. Now that the relatively good responsivity and efficient charge transfer are observed in the FACs/C16-IDTBT type-II heterojunctions (gate-voltage dependent R & PL spectral), why there is no positive shift of the threshold voltage and high photocurrent? The reason should be elaborated.

AUTHORS RESPONSE: We would like to thank the reviewer for raising this critical comment. From our spectroscopic characterisation results (**Fig. 3**), there were indeed photoholes transferred from the FACs layer to the C₁₆-IDTBT transistor channel. As pointed out by the reviewer, there only existed very limited photoresponse for the HJPT using the combination of FACs and C₁₆-IDTBT, especially in the operation regime close the transistor's off-state, whilst at the transistor's on-state a larger photoresponse was observed. The latter is nevertheless accompanied by a high dark current level. Such low photocurrent could indicate that there might be some loss(es) for those photoholes transferred out of the FACs layer. We attribute the possible loss of photoholes to the nonradiative recombination occurred at the FACs/OSC interfaces (*Adv. Mater.* 2018, 1803019). We would like to mention that there have been increasing research activities and numerous reports to tackle interfacial nonradiative losses in the field of perovskite photovoltaics (*Nature* 2019, **567**, 511–515; *Nat. Photonics* 2019, **13**, 460–466). We also note that two published works by Wang et al. (*Adv. Mater.* 2016, **28**, 6734–6739) and by Wolff et al. (*Adv. Mater.* 2017, **29**, 1700159) have showed forming an insulating layer at the interface between the perovskite and the charge transporting layer could effectively reduce the interfacial nonradiative losses. In fact, such an insulating layer forms a type-I heterojunction with the perovskite layer.

Regarding the positive shift of the threshold voltage in a phototransistor, two common mechanisms can be considered to explain this phenomenon. The first one is the change in injection barrier due to the photoelectron accumulation (also known as *photovoltaic effect*), and the second one is the change in the carrier density due to the additional electrical field induced by the trapped photocarriers (also known as *photogating effect*). For the latter, the effective field-effect induced

by the trapped holes shifts the Fermi level which, in turn, induces more electrons (*Chem. Soc. Rev.*, 2015, **44**, 3691-3718). The relatively small threshold voltage shift suggests neither of these mechanisms are pronounced in the FACs/C₁₆-IDTBT heterojunction.

Rather, the plausible explanation for such an observation is that after photogeneration and transfer of photoholes from FACs to C₁₆-IDTBT, the hole-rich C₁₆-IDTBT and electron-rich FACs are formed immediately (as evidently seen in the current-voltage characteristics for the FACs/C₁₆-IDTBT diode in **Supplementary Fig. 15** in the supplementary information of our revised manuscript). However, the FACs/OSC interfaces and/or the grain boundaries in the FACs layer could serve as the recombination sites. No apparent shift is observed in the threshold voltage for the FACs/C₁₆-IDTBT HJPT because the photoelectrons are not 'preserved' in the FACs layer. The latter is supported by our PL/TPC/TRMC characterisations (**Fig. 3**) that show the photocarrier lifetime in the FACs is much shorter in the case of FACs/C₁₆-IDTBT. On the other hand, the low photocurrent could also be attributed to a similar situation for those photoholes that are recombined via nonradiative loss routes. As such, the transferred photoholes recombine quickly before they can contribute to high photoresponse. Apart from this, we note that the mobile ionic defects in the perovskite film could also disrupt the formation of the electrostatic potential (*Chem. Soc. Rev.* 2018, **47**, 4581-4610).

In the end, we would like to point out despite that forming a type-II heterojunction between the light absorber and the transistor channel is rather intuitive, there are still many factors taking place obstructing the circulation of photocarriers in a phototransistor. We hope we have provided more insight into the case of low-photoresponse shown in the FACs/C₁₆-IDTBT HJPT. There could be several approaches to tackle this issue by borrowing the concepts that have demonstrated effective reduction in nonradiative interfacial losses from what has been reported in perovskite photovoltaics. The latter is certainly a topic worth following up. To this end, we hope the reviewer finds our response satisfactory.

REVIEWER: 6) When the authors tried to explain the role of type-I heterojunction in the phototransistor, different techniques, including PL, TRMC, Mott-Schottky analysis were performed. However, the techniques used by the authors are mainly based on the measurements of thin films or their stacks. The measurements for the full devices, e.g., the mobility of different devices, should also be provided to exhibit the impact of type-I heterojunction.

AUTHORS RESPONSE: We thank the reviewer for this highly valuable opinion on the sample characterisations. A more advanced spectroscopic measurement with an *in-situ* setup on a thin-film transistor, indeed, could resemble each operation regime in a transistor whilst revealing the underlying photophysical mechanism(s). However, for each of these characterisations there are several pre-cautions that need to be taken into account, e.g. electrical signal crosstalk. We note that studying each *in-situ* spectroscopic response in a given optoelectronic device that requires considerations in all aspects should be an individual study. To avoid any ambiguity regarding the experimental setup, in this work we have chosen to focus on the most distinguishing part in the device operation that happens to be near the turn-on voltages of our HJPTs. As such, we used the corresponding thin-film stacks that resemble each of the representative HJPTs for our spectroscopic studies whilst providing reliable characterisation results that can be confidently correlated to the device data. On the other hand, in our initial submission we have provided the key transistor performance parameters in **Supplementary Table 1** for each type of the HJPTs that were characterised in this work. We hope the reviewer finds our response satisfactory.

REVIEWER: 7) A TEM or SEM picture showing the cross-section of the devices should be provided to clearly demonstrate the device interfaces.

AUTHORS RESPONSE: We thank the reviewer for the suggestion of adding the cross-section image for identifying FACs/OSC interfaces. We have added the TEM image for the FACs/blend 1:1 film stack in **Supplementary Fig. 2** in our revised manuscript. We would like to apologise for that, despite our best efforts, we still were not able to conduct the TEM characterisation on the actual TG-BC HJPTs due to the very low conductivity of the gate dielectric layer. Nevertheless, the film stack characterised by the TEM measurement was prepared following the same processing protocol provided in our manuscript for preparing the corresponding FACs/blend 1:1 HJPT. The film thicknesses obtained for FACs and OSC (blend 1:1) are 30~50 nm and 30~40 nm, respectively (**Supplementary Fig. 2**).

Supplementary Fig. 2. | HRTEM characterisation for FACs/blend 1:1 film stack. High-resolution transmission electron microscopy (HRTEM) images for the FACs/blend 1:1 film stack deposited on ITO/glass: (a) low magnification; (b) high magnification.

REVIEWER: 8) The unit for y-axis should be added in figure 2a.

AUTHORS RESPONSE: In the revised manuscript, the y-axis label has been added to the Fig. 2a.

REVIEWER: 9) There are some typos such as “C16-BTBT” (page 4 and Figure 3b,c), etc.

AUTHORS RESPONSE: We have corrected the typos in the revised version of our manuscript.

REVIEWER: 10) The authors should carefully check the reference format, e.g., ref 19 & 25.

AUTHORS RESPONSE: We appreciate that the reviewer carefully examined our manuscript, and the reference format for these cited articles has been updated.

=====

Reviewer #2 (Remarks to the Author):

The manuscript entitled “Deciphering photocarrier dynamics for tuneable, high-performance perovskite-organic semiconductor heterojunction phototransistors” has reported a conceptual hybrid heterostructure involving metal-halid perovskite MHP (hereafter FACs) and organic small molecular film (OSCs), and found great different optoelectric response behaviors under “on-state” and “off-state”, and these optoelectric response can also be gate-tuned with various combination between FACs and blend content of C8-BTBT: C16-IDTBT. More importantly, it is very interesting to build up a photo-inverter by a strategy applying for unipolar phototransistors made by these FACs/OSCs heterojunctions. The key point of the research is to try to decipher the dynamics behind photocarriers’ charge transfer process, as well as photoresponse mechanisms in the heterostructures. However, present version of the manuscript contains too much data and puzzled explanations which make the main point unclear. Some of my suggestions and questions are listed below in order for providing to authors for their improvement.

AUTHORS RESPONSE: We thank the reviewer for the recognition of our demonstration on the concepts of FACs/OSC HJPTs and photo-inverters. As pointed out by the reviewer, the key message conveyed in this work is to understand the underlying mechanism for the photocarrier transfer and transport process between FACs and OSCs, hence the photoresponse in the HJPTs. Also we would like to apologise that in our initial submission we did not provide understandable mechanism schematics to illustrate the important findings acquired from our spectroscopic and material characterisations. We have updated the mechanism schematics (**Fig. 5**) in our revised manuscript. On the other hand, regarding the amount of the data contained in our submission, we would like to point out for the reviewer that we structure our main text in the following sequence: i) material/spectroscopic characterisations on HJPT-based film stacks (**Fig. 1**); ii) key optoelectronic properties for HJPTs (**Fig. 2**); iii) advanced material and spectroscopic characterisations for the understanding of the underlying mechanism in the HJPTs (**Fig. 3 & Fig. 4**); iv) demonstration of photo-inverter using different HJPTs to evaluate a phototransistor as well as for potential optoelectronic circuitry applications (**Fig. 6**). As such, the main text does centre the focus on as well as provide our explanation for how photocarriers behave in the proposed HJPT structures step by step from **Fig. 1** to **Fig. 4**, followed by the mechanism schematics depicted in **Fig. 5** and then an evaluation tool and optoelectronic applications using a photo-inverter configuration shown in **Fig. 6**. As for the length of the supplementary information, we have detailed our response in order to address this matter as the reviewer has brought this up in the comment #3. In short, such supporting data could be of interest to the broad readership of Nature Communications but remains optional for readers to read through in the circumstance that they feel there a need to explore more relevant information. We hope the reviewer finds our response satisfactory.

REVIEWER: 1). For all the data, the most important results perhaps are those experimental one illustrated in Figure 2(a) and schematic draw in Figure 5(a-c). However, by employing the information provided from the figure 5, I was not able to decipher the difference shown in Figure 2(a), e.g. for the large absolute value of negative bias (on-state), the drain current I_D is smaller for FACs/C8-BTBT than FACs/C16-IDTBT, in contrast the photoresponse is much larger for FACs/C8-BTBT than FACs/C16-IDTBT. For the low bias voltage (off-state), the same trend of photoresponse can be uncovered as those found under “on-state”.

To provide a reasonable explanation, the authors resorted to type-I and type-II band alignment as well as the specific structural features shown in small molecular arrangement (face-on and edge-on, ect.). However, the authors did not persuade me to believe the scenario appeared in Figure 5.

AUTHORS RESPONSE: We thank the reviewer for raising the concern regarding unclear mechanisms of our proposed phototransistor architecture. For this comment, we may further divide it into three issues:

[1.1] the lower dark current of FACs/C₈-BTBT;

[1.2] the larger photoresponse/photogain of FACs/C₈-BTBT in general when compared to FACs/C₁₆-IDTBT (covering both ON and OFF states);

[1.3] the persistently large photoresponse/photogain of FACs/C₈-BTBT that continues into the OFF state.

For [1.1], the differences in the dark current of FACs/C₈-BTBT, FACs/C₁₆-IDTBT, and FACs/blends can be understood directly from the differences in the field-effect mobility (please see **Supplementary Table 1**). Operating under no illumination, the semiconducting channel with a higher carrier mobility would yield a higher drain current (I_D). In this case, the FACs/C₈-BTBT device resulted in a lower field-effect mobility than the FACs/C₁₆-IDTBT device (which also agrees with the reported results of pristine C₈-BTBT and C₁₆-IDTBT without the FACs layer, see *Nat. Commun.* 2013, **4**, 2238, *Adv. Mater.* 2016, **28**, 7791 & *Org. Electron.* 2016, **36**, 73-81), and hence a lower dark current in the former.

As for [1.2] and [1.3], the reasons for the FACs/C₈-BTBT device having higher photoresponse or photogain can be understood from the lifetime comparison. From Eq. 3 in Methods in the main text, the photogain can be expressed as a ratio between the photocarrier lifetime and the transit time, i.e., how many times a photocarrier can circulate before recombining. The higher photogain of FACs/C₈-BTBT can be attributed to the longer carrier lifetime, which is evident from the time-resolved photoluminescence (TRPL), transient photoconductivity (TPC) and time-resolved microwave conductivity (TRMC) measurements as shown in **Fig. 3** in the main text. In other words, photocarriers in FACs/C₈-BTBT have longer lifetime and hence can contribute to more current circulation before recombining, resulting in the higher photoresponse and photogain in this device. This difference is even more pronounced in the off-state operation regime: due to the type-I heterojunction between the FACs and C₈-BTBT that prevents the charge transfer, photocarriers could reside in the FACs side, in which they have long lifetime and lead to the persistently large photoresponse. On the other hand, the charge transfer in the FACs/C₁₆-IDTBT device leads to photocarrier quenching which reduces the photoresponse as what we detailed in the response to the comment #5 from Reviewer #1.

Again, we would like to apologise for not providing clear schematics to elucidate the mechanisms behind the photocarrier transport in our initial submission, especially that we had too much mixed information in the original submitted **Fig. 5**. As such, we took the advises from the comment #1 of Reviewer #1 and the summary remark from Reviewer #3, and we dedicated our revised **Fig. 5** to illustrating the operational mechanism for our proposed, newly-established, FACs/C₈-BTBT type-I HJPT. This is also because there are already many existing reports and reviews for readers to understand how an effective type-II hybrid phototransistor work (e.g. *Adv. Mater.* 2013, **25**, 4267-4295; *ACS Photonics* 2016, **3**, 2197-2210). With this updated **Fig. 5**, we have greatly simplified the representation for the mechanism for the type-I HJPT as the latter is what this submission focuses on.

Moreover, in our responses to the comment #4 & #5 from Reviewer #1, we have detailed the differences between type-I and type-II heterojunction phototransistors using the two examples

(*Adv. Mater.* 2019, **31**, 1900763; *Adv. Mater.* 2017, **29**, 1704062) from the literature given by Reviewer #1. In particular, we pointed out in our response the existence of an interlayer or a wide-bandgap barrier could play an underestimated role to facilitate the photocarrier transport in the given examples by Reviewer #1 (*Adv. Mater.* 2019, **31**, 1900763; *Adv. Mater.* 2017, **29**, 1704062). These observations hint that such an energetic barrier is a factor that has been neglected in the field of phototransistors. Especially, we would like to point out a type-II heterojunction between the light absorber and the transistor channel is indeed intuitive for facilitating photocarrier transport. However, there might be still several factors taking place that could obstruct the circulation of photocarriers in a phototransistor as seen in what we pointed out in the first two paragraphs in this response as well as in our response to the comment #5 from Reviewer #1.

As the last point to address this insightful comment from Reviewer #2, we would like to mention that we also fabricated the FACs-only phototransistor (revised **Supplementary Fig. 17** & **Fig. R2** in this response letter) to further clarify any ambiguity regarding the role OSCs play in the photocarrier transport. The FACs-only phototransistor exhibits minimal changes in photoresponse across the entire operation range, indicating that in this case the photoresponse is weakly gate-voltage (V_G) dependent. The latter can be explained by the fact that the ions in the perovskite-only transistor could screen the applied gate bias (*J. Phys. Chem. Lett.* 2015, **6**, 3565-3571). In contrast, our HJPT approach employs the OSCs as the transistor channel in which the applied gate bias can effectively induce the channel drain current, hence resulting in V_G -dependent photodetection.

To this end, we would like to ensure the reviewer that the characterisations carried out in this work are indeed closely related to the understanding of the underlying mechanism of our HJPTs. We are deeply grateful to the reviewer for taking the time to provide this highly valuable comment. We hope the reviewer finds our response and our revised manuscript satisfactory.

Figure 5 | Operational mechanism for FACs/C₈-BTBT type-I HJPTs. Illustration of the photocarrier transport mechanism in the type-I HJPT using the FACs/C₈-BTBT film stack. The photoholes excited by light illumination can diffuse and/or accumulate at/near the FACs/C₈-BTBT interfaces and transport along the edge-on π - π stacking on the back-channel side of C₈-BTBT, resulting in the enhanced photoresponse.

REVIEWER: 2). As mentioned above, we do not know how the carrier density distributes across the interface near the heterojunction, as the authors did not even provide the thickness of FACs and OSC films. I just wonder how dependence on the heterojunction interface between the field-effect character and the thickness of OSC film?

AUTHORS RESPONSE: We thank that the reviewer to point out the necessity of supplying the information of the film thickness. As both suggested here and Reviewer #1, we have carried out the TEM characterisation on the FACs/blend 1:1 film stack (shown in **Supplementary Fig. 2**). As mentioned in our response to the comment #7 from Reviewer #1, here we would like to apologise again for that, despite our best efforts, we still were not able to conduct the TEM characterisation on the actual TG-BC HJPTs due to the very low conductivity of the gate dielectric layer. However, the FACs/blend 1:1 film stack was prepared following the same processing protocol provided in our manuscript for the preparation of the corresponding FACs/blend 1:1 HJPT. As shown in **Supplementary Fig. 2**, the film thicknesses for FACs and OSC are 30~50 nm and 30~40 nm, respectively. We hope this TEM image could provide clear information about the FACs/OSC film stack.

Supplementary Fig. 2. | HRTEM characterisation for FACs/blend 1:1 film stack. High-resolution transmission electron microscopy (HRTEM) images for the FACs/blend 1:1 film stack deposited on ITO/glass: (a) low magnification; (b) high magnification.

In fact, the well-performing HJPTs presented in our initial submission were obtained after we had numerous attempts to optimise the thickness of the perovskite layer in order to obtain functional devices. **Fig. R3** shows the representative HJPT transfer characteristics during our optimisation process. In the beginning (**Fig. R3a**) we employed an organic blend 1:3 (C₈-BTBT : C₁₆-IDTBT) that has been well-characterised in the past (*Adv. Mater.* 2016, **28**, 7791–7798) with commonly used perovskite concentrations in the range from 0.8 M to 1.4 M (e.g. using 0.8M in order to achieve similar electrical properties shown in most transistor-based reports: *Sci. Adv.* 2017, **3**, e1601935; *J. Phys. Chem. Lett.* 2015, **6**, 3565–3571; *Nat. Commun.* 2015, **6**, 8238. FACs of higher concentrations, e.g. 1.2M & 1.4M, often seen in the reports on photovoltaic applications). However, these processing conditions resulted in low electrical performance (see **Table R1** for FACs/blend 1:3). In one of our following optimisations (**Fig. R3b**), we employed a different gate dielectric material, namely AF2400 (see **Methods** in the main text, also see *Adv. Electron. Mater.*

2018, 4, 1700464), that has been shown to give better operation ranges (i.e. lower applied gate bias voltages) than the transistors based on the CYTOP gate dielectric (**Fig. R3a**; also see *Adv. Mater.* 2016, 28, 7791–7798). With the reduction of the FACs concentration down to 0.2M (**Fig. R3b**), we were able to achieve better electrical characteristics for our FACs/C₈-BTBT HJPT (see **Table R1** for FACs/C₈-BTBT). As such, we had since carried out our study in photocarrier dynamics using this optimised HJPT structure. We want to apologise for not being able to present all these pre-optimisation results as they are not the focus of this work. We hope the reviewer could understand our point and find our response satisfactory.

Fig. R3. | Transfer characteristics of HJPTs using different FACs concentrations. in a configuration of (a) FACs/blend 1:3 and (b) FACs/C₈-BTBT measured in dark.

Table R1. | Performance parameters for FACs/OSC HJPTs shown in Fig. R2.

Composition	FACs concentration (M)	Mobility (cm ² V ⁻¹ s ⁻¹)	Threshold voltage V _{TH} (V)	Turn-on voltage V _{ON} (V)	On/off ratio
FACs/blend 1:3	0.8	2.5×10 ⁻²	-20.6	-8.0	1.9×10 ⁴
	1.2	2.0×10 ⁻³	-17.6	-13.0	3.1×10 ³
	1.4	5.8×10 ⁻⁴	-28.6	-3.0	4.5×10 ²
FACs/C ₈ -BTBT	0.2	4.8×10 ⁻¹	-9.0	-1.4	3.8×10 ⁴
	1.2	2.0×10 ⁻²	-0.6	-3.5	2.9×10 ³
	1.4	1.6×10 ⁻²	-2.2	-3.5	4.8×10 ³

REVIEWER: 3). Because the picture for the illustrative photocarrier charge transfer is unclear, the vast group of characterization targeting for the photoelectric technique is logically unnecessary. Moreover, I strongly suggest not use so much supplementary notes and figures, as one paper however important can not contain so huge amount of data.

AUTHORS RESPONSE: We appreciate that the reviewer values very much the importance of the mechanism illustration in this work. We have updated **Fig. 5** in our revised manuscript. We would like to emphasise that revealing the underlying mechanism of the photocarrier dynamics not only requires the device performance characteristics (**Fig. 2**) but also needs the help from various material and spectroscopic techniques (i.e. **Fig. 3 & Fig. 4**), as well as the GIWAX characterisation (**Fig. 1f**). As a result, we were only able to depict the mechanism schematics after completed the characterisations mentioned above. These so-called “photoelectric techniques” are the key for us to rationalise the photocarriers circulation in the most efficient type-I HJPTs. We sincerely hope that our motivation to carry out these characterisations could be understood by the reviewer.

We would like to take this opportunity to thank the reviewer for suggesting us to reduce the length of our submitted supplementary information. In our current submission to Nature Communications, we do follow the guideline provided by the journal and leave all the relevant supporting data in our supplementary information file. In principle, there should not be any difficulty for readers to follow the story when reading the main text without the supplementary information. The supplementary information is also optional for readers – only if they feel there is a need to go through this part, as this will be only available from the online version of the article. In our case, the research focuses on employing different types of heterostructures to achieve on-demand high photoresponse from the proposed HJPTs as well as on the understanding of the underlying mechanism by spectroscopic and material characterisation techniques. Such information is fully covered in our main text and has been recognised by Reviewer #1 & Reviewer #3, as clearly pointed out in their summary remarks. Besides, Reviewer #2’s summary remark also reads “*The key point of the research is to try to decipher the dynamics behind photocarriers’ charge transfer process, as well as photoresponse mechanisms in the heterostructures*”. The latter is directly linked to “the focus on the understanding of the underlying mechanism by spectroscopic and materials characterisation techniques”, which is written in the main text.

On the other hand, in the supplementary information, we have included: 1) common types of phototransistors and their pros and cons; 2) optoelectronic properties of HJPTs; 3) background knowledge to support the information written in the main text. The combination of these three parts in the supplementary information ensures our manuscript to meet the need for the broad readership of Nature Communications, covering all the research fields related to this work as the best we can. To this end, although we indeed highly appreciate the reviewer’s opinion, however, we also believe that the data provided in our supplementary information provides extra support to the message conveyed in our work, especially for those readers who would like to explore relevant information. We hope the reviewer could understand our point and find our response satisfactory.

REVIEWER: 4). There were still some spelling errors or miss in the context, such as: Page 11/bottom line, Fig. 6b and 6c should be FACs/C16-IDTBT and FACs/C8-BTBT, respectively. Figure 3b of the inset should be FACs/C16-IDTBT. etc.

AUTHORS RESPONSE: We appreciate very much that the reviewer spent the time to carefully examine our manuscript, and the misspelling parts and typos have been corrected in the revised manuscript.

=====

Reviewer #3 (Remarks to the Author):

This manuscript demonstrated a hybrid heterojunction phototransistor by combining a metal halide perovskites (MHP) beneath a stack of organic semiconductor (OSC) transistor channels. By utilizing two different OSCs (C8-BTBT and C16-IDTBT) with different band structures, either “straddling-gap” or “staggered-gap” heterojunctions has been achieved. The former phototransistor exhibited state-of-the-art photogain value because of preferential edge-on molecular orientation resulted highly efficient photocarrier cycling. The optoelectronic property of phototransistors could also be tuned by changing the device between those two different types. The device concept is of novelty and this manuscript has shown systematic experimental investigation and theoretical analysis on the device with demonstration of performance enhancement. However this reviewer believes the following points should be well addressed before acceptance can be recommended:

AUTHORS RESPONSE: We are very grateful that the novelty of our HJPT concept has been recognised by the reviewer. We have addressed the reviewer’s comments point by point in our responses below and revised our manuscript accordingly. We hope the reviewer finds our revision satisfactory.

REVIEWER: (1) Can the authors provide more data on pristine perovskite based phototransistors to show how the polymers (C8-BTBT and C16-IDTBT) improve the photodetection performance?

AUTHORS RESPONSE: We thanks the reviewer to bring up this comment and would like to reason the improvement in the photoresponse using our HJPT as compared to the perovskite-only (i.e. FACs-only) phototransistor. The impact of the use of the OSC transistor channel on the photodetection performance directly reflects on the gate-voltage (V_G) dependent photoresponsivity (please see **Supplementary Fig. 17a**). The FACs-only phototransistor exhibits minimum changes in photoresponsivity across the entire V_G operation range, indicating that in this case the photoresponsivity is weakly V_G -dependent. The latter can be explained by the fact that the ions in the perovskite-only transistor channel induce a screening effect to the applied gate bias (*J. Phys. Chem. Lett.* 2015, **6**, 3565–3571). In stark contrast, our HJPT approach employs OSCs as the transistor channel, and the channel current can effectively respond to applied gate bias, hence resulting in V_G -dependent photodetection. Such V_G -dependent photoresponsivity is actually the key that separates ‘phototransistors’ from other types of photodetectors, e.g. photodiodes, as it allows phototransistors to achieve high photoresponsivity. For the latter, we have detailed the relevant discussion in our **Supplementary Note 1**.

To this end, we make a direct comparison on photoresponsivity between the FACs-only device and HJPTs. In **Supplementary Fig. 17a**, both FACs/C₈-BTBT and FACs/C₁₆-IDTBT HJPTs exhibit high photoresponsivity at the on-state that is four orders of magnitude higher than that obtained from the FACs-only device. The impact of using our HJPTs on photodetection can be realised by plotting the dark-current density dependent photogain (**Supplementary Fig. 17b**). The incorporation of OSCs in such a hybrid device allows the formation of the well-functioning transistor channel, leading to much more effective photocarrier circulation. We hope the reviewer find our response has shed light on the impact of using a perovskite light absorber and organic semiconductor transistor channels on the overall photodetection performance for phototransistors.

Supplementary Fig. 17. | Photodetection properties of FACs transistor and HJPTs. (a) Gate-voltage (V_G) dependent photoresponsivity and (b) dark-current dependent photogain measured at drain voltage $V_D = -40$ V for phototransistor based on pristine perovskite (i.e. FACs only) as well as three representative HJPTs using FACs/C₈-BTBT and FACs/C₁₆-IDTBT under the illumination at 532 nm & 500 $\mu\text{W}\cdot\text{cm}^{-2}$.

REVIEWER: (2) The authors claim that for C8-BTBT, the carriers transport through preferential edge-on molecular orientation at the MHP/OSC hetero-interfaces, therefore, leading to improved photodetection performance. The surface of perovskite may affect the device performance. Can the authors provide a clear image to show the perovskite morphology?

AUTHORS RESPONSE: We thank that the reviewer to raise the concern regarding the film morphology. Apart from the AFM information shown in **Fig 1** in the main text, we carried out the the TEM characterisation on the FACs/blend 1:1 film stack, and the acquired image is shown in **Supplementary Fig. 2**. Meanwhile, we would like to apologise for that, despite our best efforts, we still were not able to perform the TEM characterisation on the actual TG-BC HJPTs due to the very low conductivity of the gate dielectric layer. However, the characterised FACs/blend 1:1 film stack was prepared following the same processing protocol provided in our manuscript for preparing the corresponding FACs/blend 1:1 HJPT. From the TEM result shown in **Supplementary Fig. 2**, the film thicknesses for FACs and OSC are 30~50 nm and 30~40 nm, respectively. Therefore, the information obtained from the TEM characterisation shows clear FACs/OSC interfaces and could help to clarify the actual morphology of the FACs/OSC film stack. As such this addition information could provide more information that cannot be resolved by the AFM data in our original submission. We hope the reviewer finds this data useful.

Supplementary Fig. 2. | HRTEM characterisation for FACs/blend 1:1 film stack. High-resolution transmission electron microscopy (HRTEM) images for the FACs/blend 1:1 film stack deposited on ITO/glass: (a) low magnification; (b) high magnification.

REVIEWER: (3) In the introduction, the authors claim that addition of a conductive channel is inserted to increase the photoconductive gain. However, the TRMC results showed that the perovskite has higher mobility than the polymers. Can the authors explain more why they choose these two polymers to modify perovskite?

AUTHORS RESPONSE: We thank the reviewer to raise this important point regarding the mobility of a material. To obtain a proper-functioning transistor the carrier density in a transistor channel must be gate-voltage dependent (*Physics of Semiconductor Devices* by Sze and Ng). As such, the drain current can be modulated to obtain a signal amplifying effect. In analogue to this, a phototransistor tends to possess higher photoresponsivity as well as photogain than other types of photodetectors as explained in detail in our response to the reviewer's comment #1. The employed organic materials, i.e. small molecule C₈-BTBT and co-polymer C₁₆-IDTBT, are both well-known high-mobility p-type semiconductors for thin-film transistor applications. However, to achieve high field-effect mobilities in the transistors that employ these two materials, there are extra efforts required, such as adding p-dopants (e.g. *Adv. Mater.* 2016, **28**, 7791–7798; *Adv. Mater.* 2019, 1900871) and processing via special deposition techniques (e.g. *Nature* 2011, **480**, 504–508; *Nat. Mater.* 2013, **12**, 665–671; *Nat. Commun.* 2014, **5**, 3005). As mentioned in our response to the comment #3 from Reviewer #1, without any special treatment we already achieved relatively high field-effect mobility using these two organic materials (please find this information from **Supplementary Table 1** for the transistor performance parameters obtained in this work; also see *Nat. Commun.* 2013, **4**, 2238, *Adv. Mater.* 2016, **28**, 7791 & *Org. Electron.* 2016, **36**, 73-81). Most importantly, the 'mobility' obtained for our phototransistors (i.e. HJPTs in this work) that employ organic channel materials is so-called 'field-effect mobility' and based on the calculation using the gradual channel approximation (*Physics of Semiconductor Devices* by Sze and Ng). In our case, a field-effect mobility is the evaluation of carrier mobility over tens of μm (i.e. a transistor's channel length) under strong influence of an electric field-effect induced by applied gate voltages. On the other hand, the mobility obtained for FACs from the TRMC measurement is based on 'local mobility' (<nm; we note that the actual distance charges travel before they change direction depends on microwave power, frequency, carrier mobility, etc. As such, it is rather difficult to give an exact number in this case.) averaged over the illuminated sample area, under no applied DC

field. A very weak ~8-9 GHz electromagnetic field is used as the probe in these measurements (e.g. ACS Energy Lett. 2016, 1, 3, 561-565). In short, these two mobilities are fundamentally different from one another by definition. We hope the reviewer finds our explanation satisfactory.

REVIEWER: (4) By characterizing the PL and TRPL, the authors claimed that the OSC material in the blend in direct contact with FACs is mostly C16-IDTBT, which was also in line with the reported vertical phase separation. Can the author provide more straightforward evidences, such as SIMS (a surface analytical technique), to show the vertical material distribution?

AUTHORS RESPONSE: We thank that the reviewer to point out this important point, i.e. using SIMS to characterise the depth profile in order to examine if the phase separation for C₈-BTBT : C₁₆-IDTBT organic blends still preserve whilst deposited on top of a FACs perovskite layer. Following the reviewer's suggestion, we carried out the SIMS characterisation on the FACs/blend 1:1 film stack. The latter was prepared using the same processing protocol provided in **Methods**, and the result is shown in **Supplementary Fig. 9**. It is clear that in the first 10 nm, we observed a region lacking CN⁻ ions that indicates only a small fraction of (or barely any) C₁₆-IDTBT appearing in the starting period of the SIMS measurement. On the other hand, a rather larger number of counts for the CN⁻ ions are observed close to those ions (I⁻, Br⁻ and PbI⁻) that indicate the existence of the FACs perovskite layer. As such, we can confirm that the upper part of the C₈-BTBT : C₁₆-IDTBT blend is dominated by C₈-BTBT whilst the lower part of the blend is a region that C₁₆-IDTBT populates.

Supplementary Fig. 9. | Negative SIMS depth profile for FACs/blend 1:1 film stack. Secondary ion mass spectrometry (SIMS) characterisation performed for the FACs/blend 1:1 film stack.

REVIEWER: (5) The authors attributed the reason for the shorter PL decay and quenched PL signal of FACs/C16-IDTBT to more effective transfer of photocarriers from FACs to OSC, which is, however, in conflict with the band diagram shown in Figure 1c. More specifically, the higher HOMO level of C8- BTBT in theory should make the hole transfer easier than that C16-IDTBT. More discussion is need.

AUTHORS RESPONSE: We apologise that the discussion may have been unclear. In fact, the HOMO level of C₈-BTBT is ‘deeper’ (i.e. further away from the vacuum level) than that of the VB edge of FACs, and hence forming the straddling-gap as pointed out in the manuscript. We have modified the discussion pertaining to this part as follows.

‘It can be observed that the PL signal from the FACs/C8-BTBT is similar to that from the pristine FACs sample. This can be explained based on the straddling gap type-I HJ which prevents the charge transfer from the light-absorbing FACs to C8-BTBT (see Fig. 1c), i.e., photogenerated carriers are confined to the FACs layer. On the other hand, the PL signal from the FACs/C16-IDTBT sample is short-lived and strongly quenched, pointing that the photocarriers in FACs can be effectively transferred to C16-IDTBT which agrees with the staggered gap type-II HJ configuration. The TRPL and SSPL results strongly confirm our hypothesis of the HJ types formed by the two different OSCs.’

REVIEWER: (6) For the control device, except for the blend OSC material, can the authors provide one more device with layer by layer deposited C16-IDTBT and C8-BTBT, with either the former or the latter in contact with perovskite layer? Will it show similar performance with the blend one?

AUTHORS RESPONSE: We thank the referee for the useful suggestion. Indeed, studying the layer-by-layer-deposited C₁₆-IDTBT/C₈-BTBT would allow the direct performance evaluation in comparison to the spontaneous phase separation obtained from the blend. However, this experiment is extremely challenging to perform as it requires a solvent system that allows sequential deposition of two organic semiconductors that could be dissolved in similar solvent categories. Therefore, lacking solvent orthogonality not only forbids us to carry out the experiment suggested here but also requires us to initiate a research study on the deposition of layered organic semiconductors, which is out of the scope of this work as significant time and efforts are needed to fully characterise such a system. The main objective of the current manuscript is to compare the performance between type-I and type-II heterojunction phototransistors and the tuneable properties of such devices. We hope the reviewer could understand the difficulty pointed out here.

REVIEWER: (7) Some minor errors: the full name of OSC is missing, light intensity unit in Figure 6b is incorrect.

AUTHORS RESPONSE: We really appreciate that the reviewer spent the time to carefully examine our manuscript. We have provided the full name for C₈-BTBT and C₁₆-IDTBT in the revised main text page 3, i.e. 2,7-dioctyl[1]benzothieno [3,2-b][1]benzothiophene (C₈-BTBT) and indacenodithiophene-benzothiadiazole (C₁₆-IDTBT), and also we have corrected the unit in **Fig. 6b**.

Reviewers' comments:

Reviewer #1 (Remarks to the Author):

The authors have provided a response to the comments and questions. And the manuscript has been revised accordingly. I have one further question about the manuscript. In the TEM image showing the cross-section of the film stack, why the colour of OSC is dark black? OSC usually consists of light elements, which should exhibit the colour of grey or white in the TEM image. The authors should explain this.

Reviewer #2 (Remarks to the Author):

I have gone through the revised manuscript carefully, and found the latter has been much improved in expressing the research work in a more systematic and logic manner. In another word, the authors have finished showing a complete work on FACs/OSC heterojunction for a photoresponse. In this sense, the work should be published in a journal with high reputation as NC. However, the manuscript still contain unsatisfactory points (or even wrong concept) as following.

Mobility Photoresponse Gating control Carrier lifetime

FACs only O ___ X O

OSC O X O ___

FACs/OSC O O O ?

Remark: O: good;

X: poor;

___: marginal or depending

From the summary of properties for MHPs and OSC shown in the above table, we can clearly understand the merit of FACs/OSC heterojunction. However, the two key points have not been clearly explained well in the revised manuscript.

1). Comparing the properties of FACs/C8-BTBT with FACs/C16-IDBTB shown in the following table, we can find the former is much better than the latter. For the reason, the authors resort to the type-I (straddling band alignment) for FACs/C8-BTBT's better performance and argue there is no photogenerated carrier transfer from FACs to OSC. If this is true, the properties for FACs/C8-BTBT (note: the electrodes contact FACs directly) should be much similar to FACs, but the FACs has very weak photoresponse. This is remarkably contradictory.

FACs/C8-BTBT FACs/C16-IDBTB

Photoresponse best good

Transfer of photo-generated carrier no yes

Carrier life time long Short

2). The authors argued FACs/C16-IDBTB showing short carrier lifetime due to the transfer across the heterointerface owing to type-II alignment, can be attributed to the interface nonradiative recombination. But, what is the origin of the latter? And actual images are really unknown so far!

As for the wrong concept, in the Figure 5, the channel hole should be located near the interface between dielectric and OSC, not the interface between FACs and OSC.

Again, we believe some mechanisms can not be clarified completely in a single paper, therefore, we hope the authors can understand and provide a short and convince results for the present heterojunction photoresponse.

Reviewer #3 (Remarks to the Author):

This reviewer found that the authors have mostly addressed previous comments. I think the carrier dynamics has been systematically investigated and device structure/functionality has certain novelty. However there are two additional questions:

1. Although the authors have claimed several advantages of using type-I heterostructure for photodetection. Some similar work using type-II structure also based on perovskite and OSC seems to have much higher photogain, such as work here: *Light: Science & Applications* volume 6, page e17023 (2017). Can the authors further clarify the advantage of their device structure in this regard?

2. A critical question is the stability of perovskite material. As it has well-known stability issues, such as phase stability, stability in water and oxygen environment, ion migration under electric field, and light induced degradation, all these sabotage the long term stability of device. However the authors have not shown any stability test, or endurance test result. It is hard to judge the practicality of the device. Therefore can the authors provide more related and convincing data? Without these, I cannot recommend acceptance of this work.

Reviewers' comments:

=====

Reviewer #1 (Remarks to the Author):

The authors have provided a response to the comments and questions. And the manuscript has been revised accordingly. I have one further question about the manuscript. In the TEM image showing the cross-section of the film stack, why the colour of OSC is dark black? OSC usually consists of light elements, which should exhibit the colour of grey or white in the TEM image. The authors should explain this.

AUTHORS RESPONSE: We thank the reviewer for recognising our efforts to address the previous comments during the 1st round of the review process.

Regarding the TEM images, they were taken with the High-Angle Annular Dark Field Scanning Transmission Electron Microscopy (HAADF-STEM) mode of the instrument. Therefore, the contrast of the images is related to the index of elements. The organic components are darker in contrast in this case, and we have revised the following parts in the manuscript.

1. For **Supplementary Fig. 2**, the revised caption is as follows:
Supplementary Fig. 2. | HAADF-STEM characterization for FACs/blend 1:1 film stack. HAADF-STEM images for the FACs/blend 1:1 film stack deposited on ITO/glass: (a) low magnification; (b) high magnification.
2. In the Methods section, we have revised the relevant part accordingly:
High-angle annular-dark field scanning transmission electron microscopy. The sample for cross-sectional high-angle annular dark-field scanning transmission electron microscopy (HAADF-STEM) was prepared by a focused ion beam (FIB) technique using a Helios G4 UX (FEI). About 50 nm of a protective aluminium layer was deposited by a thermal evaporator (Angstrom) before performing the cross-sectional sample preparation by FIB. The HAADF-STEM image of a bilayer sample was investigated by a Titan Themis Z (FEI) transmission electron microscope operated at an acceleration voltage of 300 kV.
3. In the main text and SI, “HR-TEM” has been revised as “HAADF-STEM”.

We sincerely apologise for all the confusion caused from our side and hope the reviewer finds our revision satisfactory.

=====

Reviewer #2 (Remarks to the Author):

I have gone through the revised manuscript carefully, and found the latter has been much improved in expressing the research work in a more systematic and logic manner. In another word, the authors have finished showing a complete work on FACs/OSC heterojunction for a photoresponse. In this sense, the work should be published in a journal with high reputation as NC. However, the manuscript still contain unsatisfactory points (or even wrong concept) as following.

	Mobility	Photoresponse	Gating control	Carrier lifetime
FACs only	O	—	X	O
OSC	O	X	O	—
FACs/OSC	O	O	O	?

Remark: O: good;

X: poor;

—: marginal or depending

From the summary of properties for MHPs and OSC shown in the above table, we can clearly understand the merit of FACs/OSC heterjunction. However, the two key points have not been clearly explained well in the revised manuscript.

AUTHORS RESPONSE: We deeply appreciate that the reviewer spent the valuable time examining our manuscript with extra caution and has recognised the quality of our work and its suitability for publication in Nature Communications. The reviewer had carefully put together different combinations of FACs and OSCs used in this work in a table (please see above; reproduced by screenshotting the table from the PDF attached in the editor’s decision email) for clarifying the comments made in this round’s review. We once again thank the reviewer for all the efforts on improving the clarity of our HJPT approach.

Nevertheless, we would like to point out one part in this table that could cause different interpretation in our ‘device’ results – i.e. ‘good’ mobility for FACs. Instead, the latter should be ‘poor,’ especially in this case where we refer to the *field-effect mobility* for a TFT. To clarify the differences of mobility types we would like to bring up our response to the comment #3 by Reviewer #3 from the 1st reviewer report, which reads: “*However, the TRMC results showed that the perovskite has higher mobility than the polymers. Can the authors explain more why they choose these two polymers to modify perovskite?*”. In order to best address this point, we have attached our full response below for the reviewer’s information (in particular the underlined text).

We thank the reviewer to raise this important point regarding the mobility of a material. To obtain a proper-functioning transistor the carrier density in a transistor channel must be gate-voltage dependent (*Physics of Semiconductor Devices* by Sze and Ng). As such, the drain current can be modulated to obtain a signal amplifying effect. In analogue to this, a phototransistor tends to possess higher photoresponsivity as well as photogain than other types of photodetectors as explained in detail in our response to the reviewer’s comment #1. The employed organic materials, i.e. small molecule C₈-BTBT and co-polymer C₁₆-IDTBT, are both well-known high-mobility p-type semiconductors for thin-film transistor applications. However, to achieve high field-effect mobilities in the transistors that employ these two materials, there are extra efforts required, such as adding p-dopants (e.g. *Adv. Mater.* 2016, **28**, 7791–7798; *Adv. Mater.* 2019, 1900871) and

processing via special deposition techniques (e.g. *Nature* 2011, **480**, 504–508; *Nat. Mater.* 2013, **12**, 665–671; *Nat. Commun.* 2014, **5**, 3005). As mentioned in our response to the comment #3 from Reviewer #1, without any special treatment we already achieved relatively high field-effect mobility using these two organic materials (please find this information from **Supplementary Table 1** for the transistor performance parameters obtained in this work; also see *Nat. Commun.* 2013, **4**, 2238, *Adv. Mater.* 2016, **28**, 7791 & *Org. Electron.* 2016, **36**, 73-81). Most importantly, the ‘mobility’ obtained for our phototransistors (i.e. HJPTs in this work) that employs organic channel materials is so-called ‘field-effect mobility’ and based on the calculation using the gradual channel approximation (*Physics of Semiconductor Devices* by Sze and Ng). In our case, a field-effect mobility is the evaluation of carrier mobility over tens of μm (i.e. a transistor’s channel length) under strong influence of an electric field-effect induced by applied gate voltages. On the other hand, the mobility obtained for FACs from the TRMC measurement is based on ‘local mobility’ (<nm; we note that the actual distance charges travel before they change direction depends on microwave power, frequency, carrier mobility, etc. As such, it is rather difficult to give an exact number in this case.) averaged over the illuminated sample area, under no applied DC field. A very weak $\sim 8\text{-}9$ GHz electromagnetic field is used as the probe in these measurements (e.g. *ACS Energy Lett.* 2016, **1**, 3, 561-565). In short, these two mobilities are fundamentally different from one another by definition. We hope the reviewer finds our explanation satisfactory.

Also, the FACs and FACs/ C_8 -BTBT TFTs essentially have completely different electrical characteristics (i.e. please see **Fig. R2**, adopted from our 1st author response), and it is apparent that FACs cannot be used as the TFT channel alone as seen in our previous response letter in the 1st round review process. We have attached this specific response to the comment #2 from Reviewer #1 during the 1st reviewer process below.

Fig. R2. | Optoelectronic characterisation of HJPTs. Transfer characteristics of phototransistors with (a) FACs only and (b) FACs/ C_8 -BTBT heterojunction measured in the dark and under green illumination ($\lambda_{\text{peak}} = 525$ nm) of $705 \mu\text{W cm}^{-2}$. Both devices were operated at a constant $V_D = -40$ V. Phototransistors with FACs-only showed low I_D of $\sim 10^{-10}$ A in the dark and enhanced I_D of $>10^{-9}$ A under illumination. These results suggest that the FACs film is not capable of transporting sufficient channel current (i.e., I_D) to form a transistor. In contrast, the FACs/ C_8 -

BTBT heterojunction phototransistor exhibited i) excellent current-voltage modulation by varying gate bias voltages and ii) a significant increase in I_D at the off-state to $>10^{-7}$ A under illumination. As such, combining FACs with OSC (i.e., C₈-BTBT here) is the key to form a fully functioning phototransistor with excellent photoresponse.

The important information here, in an effort to further clarify the so-called ‘mobility’ of FACs seen in the reviewer’s summary remark table, is that the FACs/C₈-BTBT TFT is a well-functioning transistor, unlike the FACs TFT with no apparent gate modulation (**Fig. R2**). We hope we have clarified this unclear part for the reviewer. We are really grateful for the reviewer’s time and efforts to carefully analyse the large amount of information presented in our work.

REVIEWER: 1) Comparing the properties of FACs/C₈-BTBT with FACs/C₁₆-IDBTB shown in the following table, we can find the former is much better than the latter. For the reason, the authors resort to the type-I (straddling band alignment) for FACs/C₈-BTBT’s better performance and argue there is no photogenerated carrier transfer from FACs to OSC. If this is true, the properties for FACs/C₈-BTBT (note: the electrodes contact FACs directly) should be much similar to FACs, but the FACs has very week photoresponse. This is remarkably contradictory.

	FACs/C ₈ -BTBT	FACs/C ₁₆ -IDBTB
Photoresponse	best	good
Transfer of photo-generated carrier	no	yes
Carrier life time	long	Short

AUTHORS RESPONSE: We thank the reviewer to point out this important aspect. This is in fact the main advantage of HJPT that remarkably enhances the photoresponse in a TFT configuration – i.e. one of the key messages we would like to convey in our manuscript. What is important in our submission is that we are able to engineer a heterostructure to counter the poor photoresponse in the FACs TFT. We have provided the relevant discussion in our previous submissions, which reads as (the following text directly adopted from the previous revision without any modification):

To further clarify any ambiguity regarding the role OSCs play in the photocarrier transport, we processed the FACs-only phototransistor in order to shed more light on the impact of the use of the OSC transistor channel on the photodetection performance. **Supplementary Fig. 17a** compares the FACs-only device with two HJPTs using either C₈-BTBT or C₁₆-BTBT as the channel material. The FACs-only phototransistor exhibits minimal changes in photoresponse across the entire V_G operation range, indicating that in this case the photoresponse is weakly V_G -dependent. The latter can be explained by the fact that the ions in the perovskite-only transistor could screen the applied gate bias²². In contrast, our HJPT approach employs the OSCs as the transistor channel in which the applied gate bias can effectively induce the channel drain current, hence resulting in V_G -dependent photodetection. Such V_G -dependent photoresponse is in fact the key that separates ‘phototransistors’ from other types of photodetectors and allows phototransistors

to achieve high photoresponsivity, as detailed in our **Supplementary Note 1**. The impact of using our HJPTs on photodetection can also be realized by plotting the relation between photogain and dark current density (**Supplementary Fig. 17b**). In particular, the formation of FACs/C₈-BTBT type-I HJPT not only exhibits excellent channel modulation but also leads to highly effective photocarrier circulation.

We would like to remark that the aforementioned discussion is placed after our analysis on the carrier dynamics based on electrical, microscopic and spectroscopic characterisations. In summary, we have pointed out in the text that the formation of an edge-on orientation of C₈-BTBT molecules (evidenced by GIWAX shown in **Fig. 1f**) onto the FACs layer can assist the lateral charge transport in the FACs/C₈-BTBT dominated TFT channels. In addition, we would like to emphasise that, as pointed out by the Reviewer #3 as well (please read the summary remark from Reviewer #3 below), we have carried out systematic characterisations to reveal how the carrier dynamics is, and our interpretation does not solely depend on one single type of characterisations. Rather, we have put together our best efforts to synergistically combine a wide range of characterisation techniques to justify our claims. We detail this process in the next two paragraphs.

In our manuscript, we first report that we observed the good performance from both the electrical and optoelectronic characterisations (e.g. excellent current-voltage characteristics and photoresponse) for the FACs/C₈-BTBT TFTs. In the meantime, we carried out the microscopic characterisations to reveal the materials properties (i.e. an edge-on orientation). Then a series of spectroscopic characterisations were carried out. Then we learnt the majority of the photocarriers could reside with the FACs in the FACs/C₈-BTBT configuration under a no-field-effect condition in the direction perpendicular to the heterointerface. We by no means claim there is completely no charge transfer at all from FACs to C₈-BTBT as we can still observe some PL decays in **Fig. 3**. For the latter, we note that some charge transfer in a type-I heterojunction could still happen under a carrier distribution formed at the band-edge in thermal-equilibrium, which allows this to happen [*Physics of Semiconductor Devices*, Sze & Ng]. However, it is clear that for the entire transistor operation range (i.e. from the off-state to the on-state), we have seen excellent photoresponse as those photogenerated carriers do survive in the FACs/C₈-BTBT heterostructures. Even under a weak field effect for the applied voltages of $-5 \text{ V} < V_G < 0 \text{ V}$, the photocarriers could just come close to the heterointerface when operating close to the transistor's off-state. After these spectroscopic studies, we then proceeded to draw a mechanistic picture to describe this peculiar case for FACs/C₈-BTBT, in which the transport of the photocarriers could be assisted via these edge-on molecules. We note that there will be significant photocarriers transferred to the transistor channel at the on-state. On the other hand, in the circumstance of a weak field-effect induced by the gate, those photocarriers that are brought close to the transistor channel could still benefit from the preferential molecular packing for lateral charge transport along the transistor channel. This is also because we do not observe significant nonradiative recombination as evidently seen in the spectroscopic data.

We then carried out the characterisations on the FACs TFT as our reference to justify our claim and have seen no field-effect modulated current as well as poor photoresponse. We found that there are extensive and systematic studies needed if one would like to draw a full picture to elucidate the mechanism behind the HJPTs. Without taking into account the important factors, such as the preferential edge-on molecular packing in the C₈-BTBT transistor channel as well as the gate-induced field-effect, we are afraid that analysing the device performance based on the spectroscopic results might only reveal part of the working principles of the HJPTs. On the other

hand, if one considers the fact that there exists much less PL quenching in the FACs/C₈-BTBT film stack and excellent photoresponse in the FACs/C₈-BTBT TFTs, a rather intuitive explanation is that the photocarriers in the film stack of FACs/C₈-BTBT are long-lived without much nonradiative recombination like the FACs/C₁₆-IDTBT stack when those carriers are needed to transport along the organic semiconductor transistor channel. The FACs film on its own, despite that long-lived photocarriers exist, was not capable to serve as the transistor channel as evidently seen in **Fig. R2**, which is shown in our previous response to the reviewer's summary remark.

On a separate note, we do feel it is necessary to bring up our optimisation process on device architecture, for which we had detailed the work on how and what we have done for examining the best configuration to get photoresponse. The relevant information is available in our previous responses to the comments #1 and #2 from Reviewer #1 in the 1st round of the review process.

As mentioned by the reviewer in the 1st reviewer report, this is a relatively long manuscript containing a wide range of characterisations, and we have to admit it is also very challenging for us to draw a full picture without carefully assembling every single piece of information to interpret the peculiar device performance that is obtained with our HJPT approach. Especially if one were to only look at one specific characterisation, the whole picture is unlikely to be revealed in full, hence causing some inevitable misinterpretation. Thus, we sincerely hope with our response, the misunderstanding between the performance of the FACs and FACs/C₈-BTBT TFTs has been clarified for the reviewer. In the meantime, we are deeply grateful for the reviewer who spent the valuable time on examining our work with extreme caution.

REVIEWER: 2) The authors argued FACs/C₁₆-IDBTB showing short carrier lifetime due to the transfer across the heterointerface owing to type-II alignment, can be attributed to the interface nonradiative recombination. But, what is the origin of the latter? And actual images are really unknown so far!

AUTHORS RESPONSE: We thank the reviewer for providing this highly valuable comment. The shorter lifetime that we have observed from the spectroscopic characterisation has implied that there are photocarriers transferred out of the FACs layer. Thus, the most intuitive thinking is that the photocarriers have been transferred to the C₁₆-IDTBT layer due to the type-II preferential energetic alignment, which is also highlighted by Reviewer #1 in the last reviewer report. However, we have not observed an effective increase in the current level from the FACs/C₁₆-IDTBT HJPT under light illumination. As such, we have attributed the possible causes of the loss to interfacial nonradiative recombination. We have to admit this is a very challenging comment to address because currently the entire perovskite community has been working towards understanding the causes of nonradiative recombination, in particularly at the interfaces between perovskite and adjacent charge-transporting layers [*Nat. Energy* **3**, 847–854 (2018) and *Energy Environ. Sci.*, DOI: 10.1039/C9EE02020A (link) (2019)]. In these works, the content that is related to nonradiative recombination was mainly understood by observing the results from the spectroscopic data followed by direct interpretation on the difference between different structures. So does our work here, to interpret the data from our spectroscopic measurements.

Having said that we fully agree with the reviewer that understanding the origin of this loss in our FACs/C₁₆-IDTBT HJPT is very important. However, to identify the true 'origin', a further study of a significant size and a strong focus will be needed, and this is beyond the scope of the

current work. The current version of the manuscript is already 30-page long (including 32-page SI), containing a large amount of data as noted by the reviewers as well.

Nevertheless, we make an effort to suggest a future experimental plan to elucidate this particular point as follows. First of all, we need to systematically synthesise relevant molecules to understand either i) indacenodithiophene (IDT; see the molecular structure below for the red part) and/or ii) benzothiadiazole (BT; see the molecular structure below for the blue part) is responsible for this loss. By changing the copolymerization unit structure of the polymer (i.e. IDT and/or BT), we will be able to further adjust packing of synthesised polymers (e.g. synthesising polymers with either the same donor unit or the same acceptor unit; replacing the side chains [*J. Mater. Chem. A* **5**, 10798 (2017)]), morphology of resulting films, contact areas with perovskite light absorber and microstructure distribution. Apart from basic electrical and photoresponse characterisations, a range of materials and spectroscopic characterisations are required to quantitatively identify ‘the origin’ of the loss. We believe that only through a systematic study of these factors as well as by resolving this inefficient type-II architecture, we will be able to conclude the ‘origin’ of interfacial nonradiative recombination at the FACs/C₁₆-IDTBT interfaces.

We have revised our manuscript accordingly to include a brief discussion on this part as well. We certainly acknowledge the crucial importance of revealing nonradiative loss in the perovskite research and would like to contribute our knowledge to advance this field. However, we have a different focus for our current submission, and with our apologies we sincerely hope the reviewer could understand our intention to stay focused on our current device concept.

REVIEWER: 3) As for the wrong concept, in the Figure 5, the channel hole should be located near the interface between dielectric and OSC, not the interface between FACs and OSC. Again, we believe some mechanisms can not be clarified completely in a single paper, therefore, we hope the authors can understand and provide a short and convince results for the present heterojunction photoresponse.

AUTHORS RESPONSE: We would like to apologise for the not-so-perfect schematic we drew in our revised manuscript in the previous round as we found it was very challenging to present an edge-on orientation using a pure-2D schematic (i.e. the right panel in **Fig. 5** from our previous revision). We do fully agree with the reviewer’s understanding on “*some mechanisms can not be clarified completely in a single paper*” and by no means, indicate that a dominated charge transport in a TFT takes place at the FACs and OSC interface. We have revised our schematic for the

operational mechanism for the FACs/C₈-BTBT HJPT in the revised manuscript. The schematic is now drawn in a 3D presentation to replace the poor 2D drawing shown in our previous revision. We realise that the visualisation of an edge-on orientation makes a huge difference in perception, i.e. *depth of field*. Failing to show the depth of field in our last version has caused misunderstanding in how we interpret where the charge transport takes place. We had provided the relevant description for **Fig. 5** in the main text page 12 in our previous revision, and the most part remains unchanged in this revision. We apologise for the poor drawing style in our last revision and sincerely hope the reviewer finds our revision satisfactory this time.

Figure 5 | Operational mechanism for FACs/C₈-BTBT type-I HJPTs. Illustration of the photocarrier transport mechanism in the type-I HJPT using the FACs/C₈-BTBT film stack. The photogenerated holes excited by light illumination can diffuse and/or accumulate at/near the FACs/C₈-BTBT interfaces under the influence of a weak gate field-effect and transport along the C₈-BTBT transistor channel formed of organic molecules packed in an edge-on orientation, resulting in the enhanced photoresponse at/near the off-state.

=====

Reviewer #3 (Remarks to the Author):

This reviewer found that the authors have mostly addressed previous comments. I think the carrier dynamics has been systematically investigated and device structure/functionality has certain novelty. However there are two additional questions.

AUTHORS RESPONSE: We would like to thank the reviewer for very carefully examining our responses that have addressed the reviewers' comments in the previous reviewer report and for the recognition of our efforts on the investigation of charge carrier dynamics. For the extra comments in this round, we have detailed our responses below. We are highly grateful for the valuable time the reviewer spent on this work and sincerely hope the reviewer finds our responses satisfactory.

REVIEWER: 1) Although the authors have claimed several advantages of using type-I heterostructure for photodetection. Some similar work using type-II structure also based on perovskite and OSC seems to have much higher photogain, such as work here: *Light: Science & Applications* volume 6, page e17023 (2017). Can the authors further clarify the advantage of their device structure in this regard?

AUTHORS RESPONSE: We thank the reviewer for carefully examining our claims and bringing up this very important report from the literature. In fact, this work "*Light: Science & Applications* volume 6, page e17023 (2017)" has been cited and discussed in our initial submission and in our 1st revised submission (Ref. 37 in the Supplementary Information). This work is a very good example to clarify the advantage for using a proper 'semiconductor' system as what we have explained in the supplementary note "**Issues of phototransistor with highly conductive channel materials**". More specifically, in the work pointed out by the reviewer, PEDOT:PSS was employed as the channel material for the phototransistors, hence only the linear current-voltage characteristics was observed in the output characteristics in this report [i.e. *Figure 1c* in *Light: Science & Applications* volume 6, page e17023 (2017)]. To fully clarify this point, we have attached our revised explanation (SI p.5~6) below (please also read the revised **Supplementary Note 1** for the full discussion on phototransistors and photoconductors).

Issues of phototransistor with highly conductive channel materials. Based on similar hybrid-structure design as aforementioned, several different attempts that combined perovskite with other high-mobility or high-conductivity materials (e.g., carbon nanotubes^{39,40} and PEDOT:PSS³⁷) have been reported. For these approaches, high photoresponsivities $>10^4$ A W⁻¹ were achieved. Despite the promising values of photoresponsivity, devices based on such materials also induce several issues, which are not ideal for phototransistor operation. These conductor-like materials (e.g., graphene, metallic-like carbon nanotubes and PEDOT:PSS) possess high mobility at the expense of tuneability of channel conductivity⁴¹, causing poor current-voltage modulation⁴² and leading to a transistor channel that cannot be switched off. As a result, it is often observed a high level of dark current²⁹ and/or limited detectivity⁶ from these approaches. More critically, devices based on conductor-like materials usually show no saturation for channel current. The latter constrains these devices to operate in active mode only. Therefore, we would like to stress the fact that a photoconductive device based on conductor-like materials should be termed as a photoconductor rather than a phototransistor, even if this device is configured in a three-terminal structure.

Our heterojunction phototransistors (HJPTs) exhibit high photoresponsivity when using organic materials with field-effect mobilities in the range of 0.3 to $1 \text{ cm}^2 \text{ V}^{-1} \text{ s}^{-1}$. This result is particularly of interest as it demonstrates a new route towards high photoresponsivity. From **Eq. 1** (see **Methods**), photoresponsivity is defined as the ratio of photogenerated current to incident optical power. As such, increasing the level of photocurrent is the key to obtain high photoresponsivity. The concept of using high-mobility materials is to enlarge photocurrent by enhancing charge carrier circulation in a transistor channel for a given number of photogenerated charge carriers. In this scenario, devices with moderate mobility should only exhibit moderate photoresponsivity if there are the same number of photogenerated carriers involved in this circulation process. The demonstration of high values of photoresponsivity using our approach indicates that the enlargement in photocurrent is attributed to other factors, rather than enhanced photocarrier circulation through high-conductivity materials. Instead, this improvement is attributed to improved efficiency of preserving photocarriers that are involved in carrier circulation. This process is realised by engineering a type-I straddling MHP/OSC heterojunction. To enable a direct comparison between phototransistors with different mobilities, a normalised photoresponse is proposed here by calculating the quotient of photoresponsivity and mobility. The comparison of phototransistor performance of our work and other perovskite-based phototransistors is then performed by plotting the normalised photoresponse as a function of illumination intensity (**Supplementary Fig. 1**). Our work has demonstrated one of the highest normalised photoresponse values among all the PT-type phototransistors (i.e. devices capable of operating in both active mode and switch mode)^{9,17,18,24,26,33,43,44}. This value is even found to be comparable to those of gate-tuneable photoconductors^{11,30,31,34,40,45}.

In our initial submission and revision, we have also suggested using a photo-inverter to assess if a three-terminal device operates more like a phototransistor or a photoconductor. Please see the following text taken from the main text (Section: **Assessing performance metrics via a photo-inverter**, page 13) for the details.

To demonstrate this concept, a PMOS-like photo-inverter served as a common-source amplifier was fabricated using two identical FACs HJPTs, where HJPT-1 and HJPT-2 respectively, were employed as the depletion load and electrical switch controlled by the input signal (V_{IN}). In this unipolar configuration, one can directly evaluate the intrinsic amplification (A_i) of the optoelectronic components (e.g. HJPT-1 and HJPT-2) for amplifying input signals⁵³ as the latter is the main advantage for using an active electronic component like phototransistors. Moreover, A_i is an important criterion to separate a phototransistor from a photoconductor as a high A_i requires the saturation of I_D (**Supplementary Note 7** and **Supplementary Fig. 20**).

We sincerely hope the reviewer finds our response satisfactory.

REVIEWER: 2) A critical question is the stability of perovskite material. As it has well-known stability issues, such as phase stability, stability in water and oxygen environment, ion migration under electric field, and light induced degradation, all these sabotage the long term stability of device. However the authors have not shown any stability test, or endurance test result. It is hard to judge the practicality of the device. Therefore can the authors provide more related and convincing data? Without these, I cannot recommend acceptance of this work.

AUTHORS RESPONSE: We fully agree with the reviewer on this stability topic. Our authors have been actively working and publishing research works in improving perovskite stability to advance photovoltaic technologies and make commercial deployment possible in the coming years [ref: *Adv. Mater.* **29**, 1604186 (2017), *Nat. Energy* **2**, 17135 (2017) and *Nature* **571**, 245 (2019)]. A study by Domanski et al. has investigated how different environmental factors affect the cell stability [*Nat. Energy* **3**, 61 (2018)] whilst in a previously-published perspective article our authors have highlighted the longevity in perovskite optoelectronics could be better than the applications of perovskite photovoltaics as the working conditions could be less harsh (e.g. sensing light intensity far less than 1 sun) than those for solar cells [*Adv. Mater.* **29**, 1702838 (2017)].

Nevertheless, just like what the reviewer has mentioned here, a stability related test should be carried out for examining our device endurance for potential practical applications. We have performed a standard transistor bias-stress test for our FACs/C₈-BTBT HJPT at a condition of $V_G = -40$ V and $V_D = -10$ V for a continuous, non-stop period of 20,000 sec [*Adv. Electron. Mater.* **4**, 1700464 (2018)], simulating a transistor continuously operating in a working mode. We performed the current-voltage transfer characterisations at $t = 0$ and 20,000 sec, i.e. before and after the bias-stress test, in the dark and under green illumination using a LED source (525 nm) at an intensity of $705 \mu\text{W cm}^{-2}$. The reason for testing the device only at its initial and final states is because we would like to avoid the redistribution of the bias-stress-induced trapped charge carriers from the light illumination [*Adv. Electron. Mater.* **5**, 1800519 (2019); *Nat. Mater.* **14**, 193 (2015); *Sci. Adv.* **3**, e1602164 (2017)], which could result in a less-rigorous, less-severe bias-stress condition. In this extra experiment, our FACs/C₈-BTBT HJPT only exhibits minor current deterioration ($\sim 20\%$ for I_{ON} in the dark) as well as a negligible V_{ON} shift ($\Delta V_{ON} = \sim 0.1$ V), and the device has been well-functioning after the bias-stress test for this long period of bias-stressing. Our demonstration has shown an excellent bias-stress stability for a C₈-BTBT based TFT, especially compared to what has been reported in the literature [*Nat. Commun.* **5**, 3005 (2014)]. Importantly, we have only observed minor deterioration in the photocurrent level at the on-state after the bias-stress test ($\sim 21\%$ for I_{ON} under light illumination). This part now has been added to the supporting information (**Supplementary Fig. 21**) along with an additional paragraph in the main text in our latest revised manuscript. This result has demonstrated the potential for our technological approach combined with highly-stable FACs could lead to high operational stability.

We are extremely grateful that the reviewer has suggested this comment to us as the operational stability data has added highly important information to our current manuscript. We apologise for our omission in the stability topic in our previous submission. To this end, we sincerely hope the reviewer finds our response satisfactory and suggest our work for publication in *Nature Communications*.

Supplementary Fig. 21. | Bias-stress stability for FACs/C₈-BTBT HJPTs. Transfer current-voltage characteristics ($V_D = -10$ V) for a FACs/C₈-BTBT HJPT under a bias-stress stability test with a test condition of applying constant $V_G = -40$ V and $V_D = -10$ V for 2×10^4 sec: **a)** before and **b)** after the bias-stress test. The illumination was carried out using a green LED light source ($\lambda_{\text{peak}} = 525$ nm) with an intensity of $705 \mu\text{W}\cdot\text{cm}^{-2}$.

REVIEWERS' COMMENTS:

Reviewer #1 (Remarks to the Author):

The authors have carefully answered the questions and comments I raised. I think the revised manuscript can match the publication criteria of Nature Communications.

Reviewer #2 (Remarks to the Author):

The authors truly answered all questions I have concerned, so I recommend to receive the manuscript for publication.

Reviewer #3 (Remarks to the Author):

I have looked into the response and the revised manuscript carefully. Overall I think the authors have put a lot of effort to improve the clarity of the it. The two questions I raised in last round have been mostly well addressed. Although the stress test result, namely 20% change of dark current after 20,000 secs is not ideal as compared to an inorganic device, I think this result is reasonable in the field. Therefore, I have no more questions and I would recommend acceptance of this work.

Reviewers' comments:

=====

Reviewer #1 (Remarks to the Author):

The authors have carefully answered the questions and comments I raised. I think the revised manuscript can match the publication criteria of Nature Communications.

AUTHORS RESPONSE: We highly appreciate the reviewer for recommending our work to be published in Nature Communications and thank the reviewer for spending valuable time on reviewing this submission.

=====

Reviewer #2 (Remarks to the Author):

The authors truly answered all questions I have concerned, so I recommend to receive the manuscript for publication.

AUTHORS RESPONSE: We deeply appreciate the reviewer for recommending our work to be accepted for publication in Nature Communications and thank the reviewer for spending valuable time on examining our manuscript since our initial submission.

=====

Reviewer #3 (Remarks to the Author):

I have looked into the response and the revised manuscript carefully. Overall I think the authors have put a lot of effort to improve the clarity of the it. The two questions I raised in last round have been mostly well addressed. Although the stress test result, namely 20% change of dark current after 20,000 secs is not ideal as compared to an inorganic device, I think this result is reasonable in the field. Therefore, I have no more questions and I would recommend acceptance of this work.

AUTHORS RESPONSE: We highly appreciate the reviewer's final, very constructive comment on our revision from the previous round. Indeed, the bias-stability test for our FACs/C₈-BTBT HJPT has demonstrated that our approach outperforms previous reported organic transistors using the same organic semiconductor. The latter was detailed in our previous response to the reviewer. Finally, we would like to thank the reviewer for spending very valuable time on examining our work as well as for recommending publication of our manuscript in Nature Communications.